# DIFFUSION$^2$: DYNAMIC 3D CONTENT GENERATION VIA SCORE COMPOSITION OF VIDEO AND MULTI-VIEW DIFFUSION MODELS

**Zeyu Yang**[*], **Zijie Pan**,[*] **Chun Gu**, **Li Zhang**[†]
Fudan University

https://github.com/fudan-zvg/diffusion-square

## ABSTRACT

Recent advancements in 3D generation are predominantly propelled by improvements in 3D-aware image diffusion models. These models are pretrained on internet-scale image data and fine-tuned on massive 3D data, offering the capability of producing highly consistent multi-view images. However, due to the scarcity of synchronized multi-view video data, it remains challenging to adapt this paradigm to 4D generation directly. Despite that, the available video and 3D data are adequate for training video and multi-view diffusion models separately that can provide satisfactory dynamic and geometric priors respectively. To take advantage of both, this paper presents ***Diffusion***$^2$, a novel framework for dynamic 3D content creation that reconciles the knowledge about geometric consistency and temporal smoothness from these models to directly sample dense multi-view multi-frame images which can be employed to optimize continuous 4D representation. Specifically, we design a simple yet effective denoising strategy via score composition of pretrained video and multi-view diffusion models based on the probability structure of the target image array. To alleviate the potential conflicts between two heterogeneous scores, we further introduce variance-reducing sampling via interpolated steps, facilitating smooth and stable generation. Owing to the high parallelism of the proposed image generation process and the efficiency of the modern 4D reconstruction pipeline, our framework can generate 4D content within few minutes. Notably, our method circumvents the reliance on expensive and hard-to-scale 4D data, thereby having the potential to benefit from the scaling of the foundation video and multi-view diffusion models. Extensive experiments demonstrate the efficacy of our proposed framework in generating highly seamless and consistent 4D assets under various types of conditions.

## 1 INTRODUCTION

Spurred by the advances from generative image models (Ho et al., 2020; Song et al., 2021a;b; Karras et al., 2022; Zhang et al., 2023), automatic 3D content creation (Poole et al., 2023; Wang et al., 2023; Tang et al., 2024b; Hong et al., 2024) has witnessed remarkable progress in terms of efficiency, fidelity, diversity, and controllability. Coupled with the breakthroughs in 4D representation (Yang et al., 2024; Wu et al., 2024a; Duan et al., 2024; Li et al., 2024), these advances further foster substantial development in dynamic 3D content generation (Singer et al., 2023; Bahmani et al., 2024; Jiang et al., 2024b; Zhao et al., 2023; Ren et al., 2023; Gao et al., 2024), which holds significant value across a wide range of applications in animation, film, game, and MetaVerse.

Recently, 3D content generation has achieved considerable breakthroughs in efficiency. Some works (Liu et al., 2023; 2024; Shi et al., 2023; Wang & Shi, 2023; Tang et al., 2024c) inject stereo knowledge into the image generation model, enabling these 3D-aware image generators to

---

[*]Equally contributed

[†]Li Zhang (lizhangfd@fudan.edu.cn) is the corresponding author with School of Data Science, Fudan University.

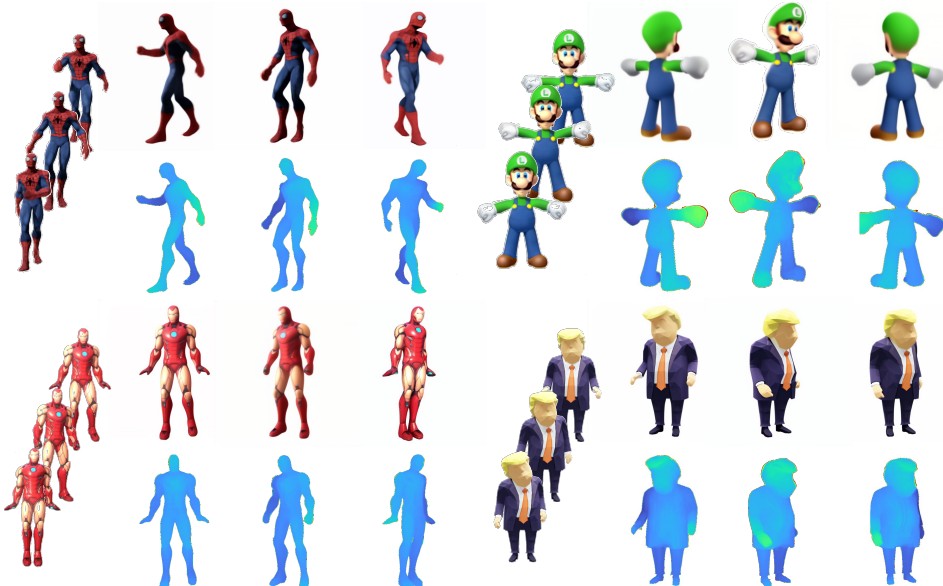

Figure 1: **Diffusion**$^2$ is designed to generate dynamic 3D content by creating a dense multi-frame multi-view image matrix in a highly parallel denoising process by combining the foundation video and multi-view diffusion model. Please refer to our *supplementary demo video* for more results.

produce consistent multi-view images, thereby effectively stabilizing and accelerating the optimization. Other efforts (Hong et al., 2024; Chen et al., 2024; Voleti et al., 2024; Zuo et al., 2024; Tang et al., 2024a) attempt to directly generate 3D representations, such as triplane (Chen et al., 2022) or 3DGS. However, the efficiency improvement from these works is largely data-driven (Wu et al., 2023; Reizenstein et al., 2021; Yu et al., 2023; Deitke et al., 2023). Consequently, it is infeasible to adapt them to 4D generation due to the scarcity of synchronized multi-view video data. Therefore, most existing 4D generation works (Jiang et al., 2024b; Yin et al., 2023; Ren et al., 2023) still adopt the score distillation sampling (SDS) approach and suffer from slow or unstable optimization.

However, despite the paucity of 4D data, there are vast available monocular video data and static multi-view image data. Existing works have demonstrated that it is feasible to train diffusion-based generative models learning the distribution of these two classes of data independently (Voleti et al., 2024; Liu et al., 2024; Tang et al., 2024c; Blattmann et al., 2023b;a). Considering that video diffusion model stores the prior of motion and temporal smoothness, and multi-view diffusion model has sound knowledge of geometrical consistency, combining the two types of generative models to generate 4D assets becomes a highly promising and appealing approach.

To leverage both of them, we propose a novel 4D generation framework, *Diffusion*$^2$, which reconciles the video and multi-view diffusion priors to directly sample multi-frame multi-view image arrays imitating the photographing process of 4D object. Instead of unleashing the pre-trained knowledge stored in model parameters through fine-tuning (Xie et al., 2024; Ren et al., 2024), *Diffusion*$^2$ achieves this goal in a training-free and architecture-agnostic manner. Our key insight is that the score function for the joint distribution of elements in image array can be approximated by the convex combination of scores for each view and frame due to the conditional independence within the image matrix. Therefore, we can directly sample multi-view multi-frame images within the framework of score-based generative models with off-the-shelf score estimators.

In practice, due to the potential divergence between the learned distributions of two foundational models caused by imbalanced training data, conflicts could emerge during the denoising process. Such conflicts are primarily manifested as two distinct modes in the denoising results, which may compromise the completeness and consistency of the generated images with continuously reducing noise levels. However, we found that this issue can be effectively resolved by leveraging the inherent properties of the diffusion framework, without requiring any additional training. Based on this idea, we further propose the variance-reducing sampling (VRS), which can effectively mitigate such conflicts at the cost of negligible additional time consumption by reconciling the distinct modes at a higher noise level. With this technique, our *Diffusion*$^2$ can produce smooth multi-view videos with

high spatial-temporal consistency. Furthermore, we introduce a dynamic and texture-decoupled reconstruction strategy that transforms the generated discrete image matrix into a continuous 4D representation while minimizing the potential detail loss caused by aggressive variance reduction, especially on images that are far from the reference view. Thanks to the highly parallel denoising in the image matrix synthesis stage and the efficiency of the reconstruction stage, our method can generate high-fidelity and diverse 4D assets within just a few minutes.

Our contributions can be summarized as follows: **(i)** We develop a novel 4D generation framework that can generate highly consistent multi-view videos in a single pass of reverse diffusion process using only existing multi-view and video diffusion models, without relying on any 4D dataset. The core of this framework is to estimate the score function of the image matrix by a convex combination of the scores predicted by orthogonal diffusion models. We identify the conditional independence within elements constituting the image arrays and establish the theoretical soundness of the proposed approaches based on this property. **(ii)** We propose the VRS for reconciling heterogeneous scores during denoising, which can effectively alleviate potential conflicts and facilitate the generation of seamless results with high consistency across multiple views and frames. In synergy with it, we also propose a dynamic and texture decoupling reconstruction pipeline to better preserve texture details. **(iii)** Systematic experiments demonstrate that our proposed method achieves impressive results under different types of prompts. **(iv)** Notably, our work first proves that it is practical to directly sample highly spatial-temporal consistent multi-view videos of dynamic 3D objects within a single-pass of denoising diffusion procedure, while only using the existing multi-view and video diffusion prior without any additional training on expensive and hard-to-scale 4D dataset.

## 2 RELATED WORK

**3D generation**    3D generation aims at creating static 3D content from different type of conditions like text or reference image. Early efforts employed GAN-based approaches (Gao et al., 2022; Schwarz et al., 2020). Recently, significant breakthroughs have been achieved with diffusion models (Ho et al., 2020). DreamFusion (Poole et al., 2023) introduced score distillation sampling (SDS) to unleash the creativity in diffusion models. A series of subsequent works (Wang et al., 2023; Shi et al., 2023; Wang & Shi, 2023; Pan et al., 2024a; Tang et al., 2024b; Yi et al., 2024) continuously address challenges such as multi-face Janus issues and slow optimization. On the other hand, with the development of large scale 3D datasets (Yu et al., 2023; Deitke et al., 2023), many works try to directly generate 3D contents by using diffusion models. Some studies (Nichol et al., 2022; Jun & Nichol, 2023; Hong et al., 2024; Tang et al., 2024a; Wang et al., 2024) have explored the direct generation of 3D representations. Another line of research (Liu et al., 2023; 2024; Long et al., 2023; Chen et al., 2024; Tang et al., 2024c) focuses on generating multi-view images with sufficient 3D consistency for reconstruction. Our method also directly generates consistent images for reconstruction. But unlike 3D counterparts, there is no large-scale synchronized multi-view video data. Therefore, we opt to combine existing geometrical consistency priors and video dynamic priors.

**Video generation**    Recent diffusion-based video generation methods have exhibited unprecedented levels of realism, diversity, and controllability. Video LDM (Blattmann et al., 2023b) is the pioneer work to apply the latent diffusion framework (Rombach et al., 2022) to video generation. The subsequent work SVD (Blattmann et al., 2023a) followed its architecture and made effective improvements to the training recipe. W.A.L.T (Gupta et al., 2023) employed a transformer with window attention tailored for spatiotemporal generative modeling to generate high-resolution videos. VDT (Lu et al., 2024) introduced the video diffusion transformer and a spatial-temporal mask mechanism to flexibly capture long-distance spatiotemporal context. The recently introduced SORA (Brooks et al., 2024) demonstrated a remarkable capability to generate long videos with intuitively physical fidelity. Models trained on large-scale video data can synthesize realistic dynamics. Besides, video diffusion models can also provide effective prior for 3D generators (Chen et al., 2024; Han et al., 2024; Voleti et al., 2024). Therefore, we build our method on this flourishing domain.

**4D generation**    Animating category-agnostic objects is challenging and has drawn considerable attention from both academia and industry. Unlike 3D generation, 4D generation requires both consistent geometry and realistic dynamics. Recent works on this domain can be categorized based on the input condition. Some of them create 4D content from text or single image. For instance, MAV3D (Singer et al., 2023) first employs SDS in text-to-4D tasks. 4D-fy (Bahmani et al., 2024)

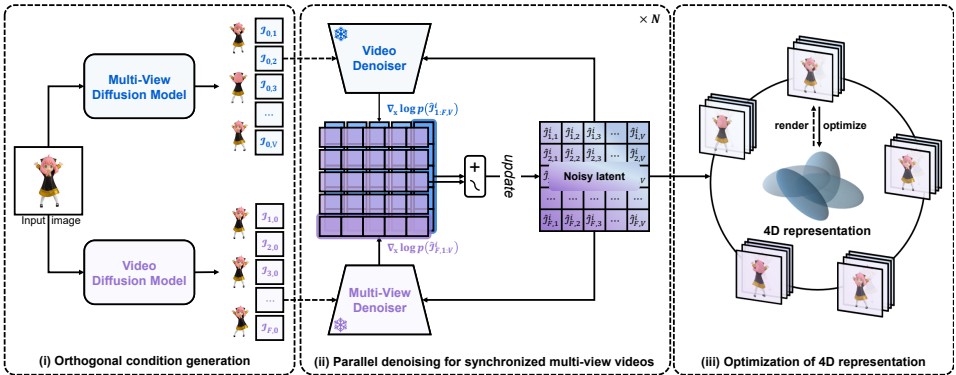

Figure 2: The overall pipeline of *Diffusion*$^2$. **(i)** Given a reference image, *Diffusion*$^2$ first independently generates the animation under the reference view (denoted $\mathcal{I}_{1:V,0}$) and the multi-view images at the reference time (denoted $\mathcal{I}_{0,1:F}$) as the condition for the subsequent generation of the full matrix, denoted $\mathcal{I}$. Depending on the type of given prompt, the condition images $\mathcal{I}_{1:V,0}$ or $\mathcal{I}_{0,1:F}$ can be specified by users. **(ii)** Then, *Diffusion*$^2$ directly samples a dense multi-frame multi-view image array by blending the estimated scores with a weighting scheduler from pretrained video and multi-view diffusion models in the reverse diffusion process. **(iii)** The generated image arrays are employed as supervision to optimize a continuous 4D content representation.

hybridizes different diffusion priors during SDS training. But they suffer from extremely slow generation. DreamGaussian4D (Ren et al., 2023) adopts deformable 3D Gaussian (Yang et al., 2024) as the underlying 4D representation and exports mesh for texture refinement, significantly improving efficiency. Another line of work (Jiang et al., 2024b; Wu et al., 2024b; Yin et al., 2023; Pan et al., 2024b) predicts dynamic objects from a single-view video with largely dictated motion. Consistent4D (Jiang et al., 2024b) proposes to use SDS approach for geometry consistency and frame interpolation loss for temporal continuity. 4DGen (Yin et al., 2023) further grounds the 4D content creation with pseudo labels. Meanwhile, L4GM (Ren et al., 2024) directly predicts consistent 3D Gaussians for each frame. And Jiang et al. (2024a) animates existing 3D assets given their multi-view rendering with 4D-SDS. Efficient4D (Pan et al., 2024b) mimics a photogrammetry-based neural volumetric video reconstruction pipeline by directly generating multi-view videos for reconstruction. Some recent works opt for training diffusion models to generate monocular video (Liang et al., 2024) or multi-view videos (Xie et al., 2024) with spatiotemporal consistency for subsequent explicit reconstruction. Compared to previous works, our framework can efficiently generate diverse 4D contents from different input prompts, avoiding the slow and unstable optimization and have potential to continuously benefit from the scalability of underlying diffusion models.

# 3 METHOD

As depicted in figure 2, the proposed 4D generation framework **_Diffusion_**$^2$ adopts a two-stage pipeline: dense observation synthesis followed by reconstruction. In this section, we will first elaborate on how to generate dense multi-view multi-frame images for reconstruction through a highly parallelizable denoising process by reconciling the pre-trained video diffusion prior and multi-view diffusion prior while pointing out why it is feasible (stage-1), and then briefly introduce how to robustly reconstruct 4D content from the sampled images (stage-2).

## 3.1 ESTIMATING SCORES OF THE IMAGE MATRIX

In this stage, our goal is to generate highly consistent dense multi-frame multi-view images for reconstruction, which can be denoted as a matrix of images:

$$\mathcal{I} = \left\{ I_{i,j} \in \mathbb{R}^{H \times W \times 3} \right\}_{i=1,j=1}^{V,F} = \begin{bmatrix} I_{1,1} & \cdots & I_{1,j} & \cdots & I_{1,F} \\ \vdots & \ddots & \vdots & \ddots & \vdots \\ I_{i,1} & \cdots & I_{i,j} & \cdots & I_{i,F} \\ \vdots & \ddots & \vdots & \ddots & \vdots \\ I_{V,1} & \cdots & I_{V,j} & \cdots & I_{V,F} \end{bmatrix}, \tag{1}$$

where $V$ is the number of views, $F$ is the number of video frames, and $(H, W)$ is the size of images. We aim to construct a generative model that allows us to directly sample natural $\mathcal{I} \sim p(\mathcal{I})$.

Now, let us first divert our focus to reviewing existing diffusion-based generators for video and multi-view images, which can be utilized for sampling realistic image sequence through the probabilistic flow ODE as below:

$$dx = -\dot{\sigma}(t)\sigma(t)\nabla_x \log p\left(x; \sigma(t)\right) dt. \qquad (2)$$

Here, $x = \left\{ I_i \in \mathbb{R}^{H \times W \times 3} \right\}_{i=1}^N$ is a series of images with $N$ frames or $N$ views, $\nabla_x \log p\left(x; \sigma\right)$ is the score function, which can be estimated as $\nabla_x \log p\left(x; \sigma\right) \approx \left(D_\theta(x; \sigma)\right)/\sigma^2$ (Karras et al., 2022; Blattmann et al., 2023a), where $D_\theta(x; \sigma)$ is a neural network trained via denoising score matching. We want to extend the above formulation to the generation of the whole image matrix, *i.e.*, sampling $\mathcal{I}$ through equation 2 according to its score function $\nabla \log p\left(\mathcal{I}; \sigma(t)\right)$. The question is, how do we estimate the score of the joint distribution of $V \times F$ images? Due to the scarcity of available synchronized multi-view videos and the potentially huge memory footprint of simultaneously consuming $V \times F$ images, it is impractical to train a neural network to directly predict the score function of an image matrix with densely distributed views and sufficient frames for substantial motion.

Fortunately, image matrix $\mathcal{I}$ in nature has a nice property, such that we can approximate the $\nabla \log p\left(\mathcal{I}; \sigma(t)\right)$ by combining existing video and multi-view score estimators, as claimed in the following theorem:

**Theorem 3.1.** *For* $x = I_{i,j}$, *we have*

$$\nabla_x \log p(\mathcal{I}; \sigma(t)) = \nabla_x \log p(\mathcal{I}_{\{1:V\},j}; \sigma(t)) + \nabla_x \log p(\mathcal{I}_{i,\{1:F\}}; \sigma(t)) - \nabla_x \log p(I_{i,j}; \sigma(t)). \quad (3)$$

Here $\nabla_x \log p(\mathcal{I}_{i,\{1:F\}})$ and $\nabla_x \log p(\mathcal{I}_{\{1:V\},j})$ can be predicted by the off-the-shelf video diffusion model and the multi-view diffusion model. This assertion implies that we can sample the desired image matrix by progressively denoising from pure Gaussian noise using the summation of two estimated scores for its row and column, which can be obtained from the pre-trained multi-view and video diffusion models respectively. The theorem 3.1 can be derived from the following assumption about the structure of $p(\mathcal{I})$ (refer to appendix A.3 for detailed proof):

**Assumption 3.1.** *images captured from different viewpoints and different times are conditionally independent given the image at the intersection of their respective rows and columns i.e.,*

$$p\left(I_{i',j}, I_{i,j'} | I_{i,j}\right) = p\left(I_{i',j} | I_{i,j}\right) p\left(I_{i,j'} | I_{i,j}\right) \quad \text{with } i' \neq i, j' \neq j. \qquad (4)$$

We provide an example in figure 3 to illustrate this assumption. Denote $I_{i,j}$ as the front view observed currently, it can indeed dictate some aspects of the back view at the same frame $I_{i',j}$, such as its contour. The conditional independence means that the future front view $I_{i,j'}$ cannot provide more information about the $I_{i',j}$ beyond those have been provided by the current front view. It aligns with our intuition, as the differences between $I_{i,j'}$ and $I_{i,j}$ are mainly induced by the hand waving, which is incapable of bringing extra information gains about the back view. Similarly, the texture of the back at the current moment is also not helpful in inferring the future dynamics.

Actually, the assumption of conditional independence is also implicitly held by many other 4D generation pipelines. For instance, the legitimacy of independently synthesizing multi-view reference and driving video (Yin et al., 2023; Xie et al., 2024) from a single image also relies on this hypothesis.

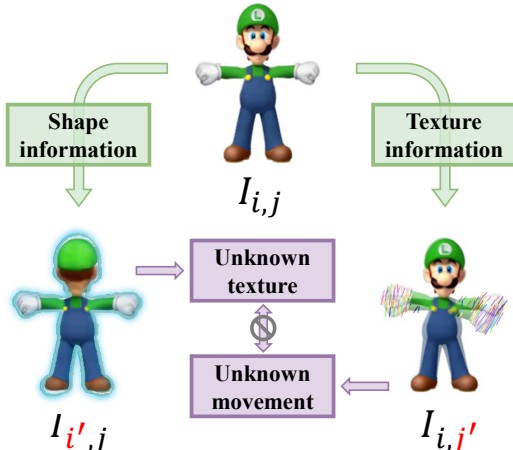

Figure 3: **Schematic illustration for the conditional independence.** Compared to $I_{i,j}$, the additional information (future dynamic) in $I_{i,j'}$ does not help in inferring the unknown texture of the back view $I_{i',j}$, and vice versa.

**Sampling in latent space** For simplicity, the previous discussion assumes that we sample images directly in the RGB space. However, modern high-resolution diffusion models typically generate images in a latent space encoded by pre-trained Auto-Encoder (Kingma & Welling, 2014; Van Den Oord et al., 2017). The legitimacy of the aforementioned derivation requires that the multi-view and video generative models share the same latent space. Therefore, we employ SV3D (Voleti et al., 2024) and SVD (Blattmann et al., 2023a) as the respective generators in this work. We believe that the requirement will be increasingly satisfied by more models in the future. As pointed out by Blattmann et al. (2023a); Voleti et al. (2024); Chen et al. (2024), video generative models trained on large-scale video datasets have learned a strong geometry prior and thus capable of providing a better pre-training for multi-view diffusion models than those trained solely on image data.

## 3.2 GENERATION UNDER VARIOUS CONDITIONS

Note that the formulation described above is based on unconditional generation. However, we are more interested in controllable generation in practice. In this section, we will discuss on how to extend the above process to the conditional generation.

First of all, since the $\nabla_{\mathrm{x}} \log p(I_{i,j})$ is intractable in the conditional generation, we use the convex combination of $\nabla_{\mathrm{x}} \log p(\mathcal{I}_{i,\{1:F\}})$ and $\nabla_{\mathrm{x}} \log p(\mathcal{I}_{\{1:V\},j})$ to replace it, and revise the equation 3 as:

$$\nabla_{\mathrm{x}} \log p\left(\mathcal{I};\sigma(t)\right) = s\nabla_{\mathrm{x}} \log p(\mathcal{I}_{i,\{1:F\}};\sigma(t)) + (1-s)\nabla_{\mathrm{x}} \log p(\mathcal{I}_{\{1:V\},j};\sigma(t)). \tag{5}$$

Then we formulating our conditional generation pipeline like a "inpainting" process for the augmented matrix $\mathcal{I}_{\mathrm{aug}}$ defined as

$$\mathcal{I}_{\mathrm{aug}} = \begin{bmatrix} I_{0,0} & \mathcal{I}_{0,\{1:V\}} \\ \mathcal{I}_{\{1:F\},0} & \mathcal{I} \end{bmatrix}. \tag{6}$$

Compared to the unconditional generation, the augmented matrix $\mathcal{I}_{\mathrm{aug}}$ includes the optional inputs $I_{0,0}, \mathcal{I}_{0,\{1:V\}}$ and $\mathcal{I}_{\{1:F\},0}$ representing reference geometry and dynamic of target object.

In the proposed generation pipeline, $\mathcal{I}_{0,\{1:V\}}, \mathcal{I}_{\{1:F\},0}$ should be first created as the known part according to the given input. Then the rest part of $\mathcal{I}_{\mathrm{aug}}$ will be sampled given these orthogonal conditions. To incorporate them into the generation of $\mathcal{I}$, we condition the score estimator for denoising each row/column in equation 5 on the first element of the same row/column in $\mathcal{I}_{\mathrm{aug}}$. This formulation enables us to handle various 4D generation tasks depending on different forms of conditions specified by users. For example, in the image-to-4D task, given a single image $I_{0,0}$ as input, both $\mathcal{I}_{0,\{1:V\}}$ and $\mathcal{I}_{\{1:F\},0}$ should be generated by multi-view and video diffusion models respectively. In the video-to-4D task, we just need to leave the input single-view video as $\mathcal{I}_{\{0:F\},0}$, and use its last frame $I_{0,0}$ as the condition for multi-view diffusion model to generate $\mathcal{I}_{0,\{1:V\}}$. These process can also be similarly extended to the text-to-4D tasks and end-to-end animating static 3D objects. Once $\mathcal{I}_{0,\{1:V\}}, \mathcal{I}_{\{1:F\},0}$ are obtained, we can sample the full matrix $\mathcal{I}$ using equations (3) and (5).

**Parallel denoising** Assumption 3.1 ensures the safety of independently generating the geometry $\mathcal{I}_{0,\{1:V\}}$ and the motion $\mathcal{I}_{\{1:F\},0}$. Since these two generation processes have no computational or data dependency, their total time cost could be reduced to a single reverse diffusion process. Additionally, when we denoise the rest part of $\mathcal{I}_{\mathrm{aug}}$, *i.e.*, $\mathcal{I}$, the score estimation for each row and column can also be parallelized in each denoising step. Therefore, with sufficient GPU memory, the total time spent figure 2 (ii) remains the same as that for generating a single video.

## 3.3 RECONCILING JOINT DENOISING

In practice, the training data for foundational multi-view and video generation models may exhibit a domain gap and imbalance in quantity; for instance, the multi-view diffusion model employed in this work is fine-tuned on an object-centric dataset which is much smaller than the large-scale general video dataset used to train the video diffusion model. Consequently, the learned distributions may not aligned ideally, leading to potential conflicts when directly fusing their scores without taking some measures to harmonize their denoising trajectories. A straightforward solution is to fine-tune the utilized video diffusion model on single-view videos of dynamic objects. However, such conflicts can also be mitigated by harnessing the capabilities inherent in the diffusion framework without extra training. Consequently, to preserve our training-free merit and avoid reliance on multi-view video data, we propose the following inference-time approach to reconcile them.

**Variance-reducing sampling via interpolated steps**  Due to the potential conflicts between the heterogeneous scores mentioned above, $I^t$ may not always consistently conform to the ideal distribution of $p(\mathcal{I}, \sigma(t))$. Obvious flicker and ghosting artifacts will appear with the accumulation of the divergence step by step. Therefore, we proposed an ingenious strategy to resolve this issue, which involves incorporating some "intermediate" steps into the reverse diffusion process. Specifically, we observe that although $\mathcal{I}^t$ may deviate from $p(\mathcal{I}, \sigma(t))$, it still possesses a high likelihood at a larger noise level. This can be understood from the following perspective: deviations between $\mathcal{I}^t$ and $p(\mathcal{I}, \sigma(t))$ are primarily exhibited as two isolated modes driven by independent score estimators. However, these can be mixed in a mollified distribution with a higher noise level, which is actually one of the initial motivations of the diffusion model (Song & Ermon, 2019). Therefore, the core insight of the proposed strategy is to deceive the diffusion model to denoise a noisy latent with a smaller variance at a higher noise level. To balance efficiency and quality, we achieve this goal using the following approach. In the $t$-th denoising step, we first update $\mathcal{I}^t$ to $\mathcal{I}^{t+1}$ based on the estimated score from equation 5. Then, we run a forward process to resample a noisy latent at noise levels between $t$ and $t + 1$. We then use it as $\mathcal{I}^t$ to repeat the previous denoising step. Algorithm 1 provides the pseudo-code for the sampling process. Since the divergence primarily occurs at the early stage of denoising, we only need to perform the rollback in the first five denoising steps, which only brings about $1/10$ extra inference steps, and the experiments demonstrate that it is sufficient to achieve very seamless and consistent results.

## 3.4 Reconstruction from sampled image matrix

Given the multi-view videos synthesized in stage-1, it is imperative to transform them into a continuous 4D representation for facilitating real-world applications. Beyond the functional considerations, *i.e.*, enabling renderings from arbitrary viewpoints beyond the images generated in stage-1, the reconstruction phase (stage-2) is also responsible for enhancing texture details in views that are distant from the reference views and frames, since we have sacrificed some texture diversity on these views with higher uncertainty with the aggressive variance-reducing strategy in stage-1 for efficiency. Therefore, we propose a geometry and texture decoupled scheme to train dynamic Gaussians on the images generated in stage-1 to minimize the negative impact of this trade-off.

**4D Gaussian optimization**  Specifically, we represent dynamic objects as a set of Gaussian primitives in canonical space, driven by a learnable deformation field. The deformation field is responsible for predicting the 6-DoF transformation of each Gaussian given its mean and the queried timestamps. We model it with an MLP (Yang et al., 2024) because of its inherent inductive bias about low-frequency and topological invariance which can effectively regularize the learning of dynamics.

In the optimization, during the initial 3,000 warm-up iterations, we only optimize the underlying static Gaussians using synthetic multi-view images at the reference frame, *i.e.*, $\mathcal{I}_{\{:,F\}}^t$, with photometric losses. Compared to warm-up with monocular views (Ren et al., 2023), consistent multi-view static images allow for rapidly achieving very high-quality geometry in this stage and bind the reference frame to the canonical space to some extent, thus making it easier for the subsequent potential extraction of deformed mesh. After warm-up, we use the full image matrix as ground truth to optimize the deformation field while fine-tuning the canonical Gaussians at a reduced learning rate and loss scale away from the reference view and frame. Unlike Ren et al. (2023) optimizing the deformation field with a single-view driving video, our consistent multi-view videos can lead to more stable optimization. Benefiting from the strong fitting capabilities of the Gaussian approach, the trained Gaussian model can achieve high-fidelity details while supporting real-time viewing.

## 4 Experiments

### 4.1 Implementation details

In the first stage, we employ Stable Video Diffusion (Blattmann et al., 2023a) as our foundation video diffusion model, predicting 25 frames each time conditioning on the reference image. SV3D$^p$ (Voleti et al., 2024) is chosen as our foundation multi-view diffusion model. For simplicity, we only generate orbital videos with 21 uniformly spaced azimuths and fixed elevation. By default, we set the number of sampling steps to 50 for both models. In the reconstruction stage, we assume

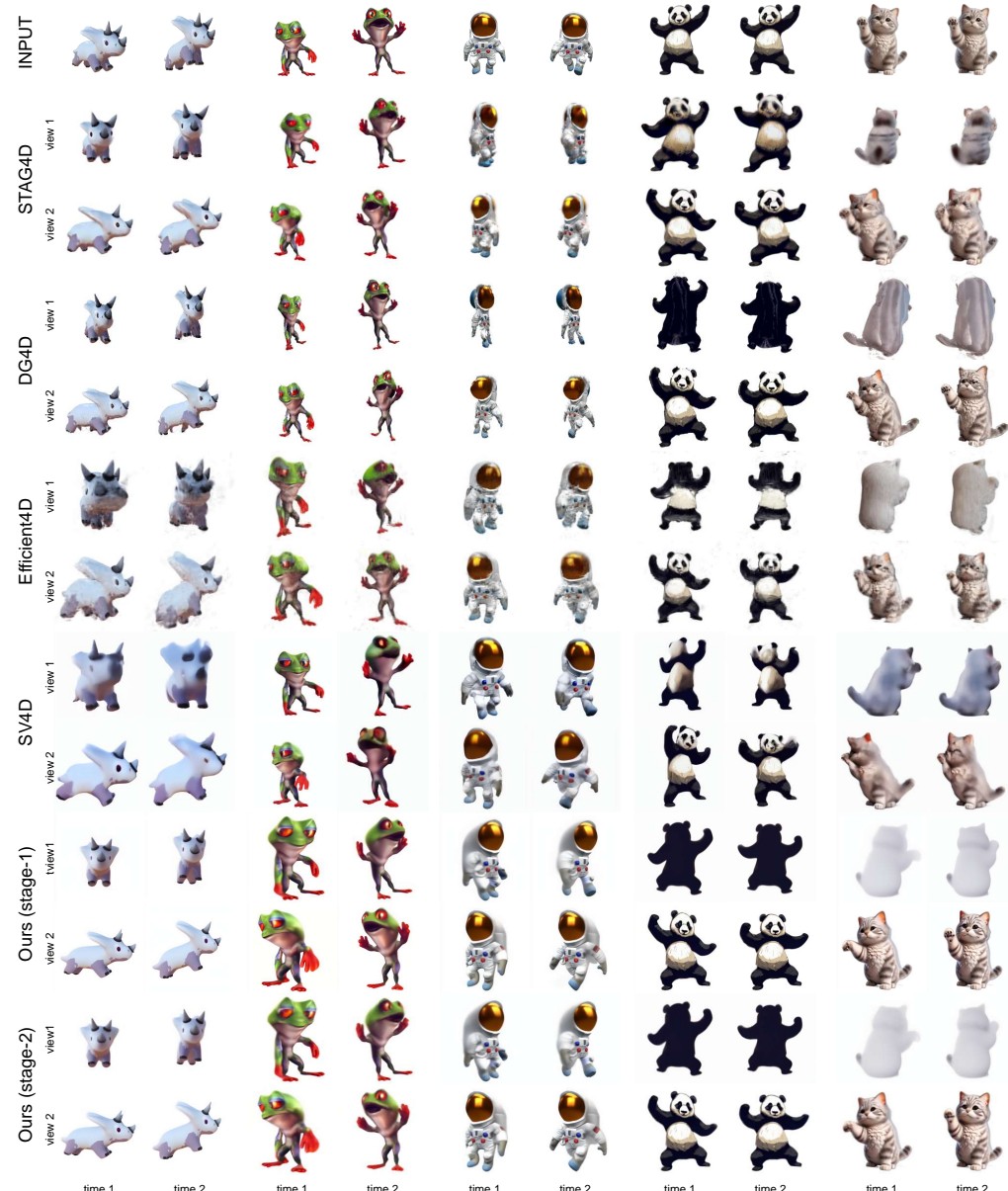

Figure 4: **Qualitative comparisons.** The left 3 cases are generated from input videos, whilst the right 2 cases are generated given single input images. The results of our both stages exhibit better coherence, appear more natural, and have fewer artifacts. More results in *supplementary video*.

that the virtual camera whose FOV is set to 33.8° orbiting around the object center with a fixed radius of $2m$. Finally, We optimized Gaussian Splatting in totally 5,000 iterations in our reconstruction stage. The image size is set to ($576 \times 576$) in both stages. All the experiments are conducted on 8 NVIDIA A6000 GPUs. Please refer to section A.5 for more details.

## 4.2 QUALITATIVE COMPARISONS

To demonstrate the effectiveness of the proposed method, we qualitatively compared our method with three representative baselines, ranging across optimization-based, photogrammetry-based, different types of diffusion prior and 4D representation. The results on the two commonly used 4D generation task, *i.e.*, video-to-4D and image-to-4D task are shown in figure 4. For fair comparison, we use the same reference video for all methods in the image-to-4d task. In terms of generation quality, our simple and elegant pipeline is capable of generating assets with remarkably superior

Table 1: **User study** on image-to-4D generation. The proportions of different methods that best match user preferences under four criteria are reported.

| | Consistent4D | STAG4D | DG4D | Efficient4D | Ours |
|---|---|---|---|---|---|
| **Ref. consistency** | 3.1% | 5.5% | 2.0% | 1.1% | **88.3%** |
| **Multi-view consistency** | 6.7% | 22.9% | 3.1% | 8.4% | **58.9%** |
| **Temporal smoothness** | 8.0% | 36.2% | 2.0% | 10.7% | **43.1%** |
| **Overall model quality** | 8.2% | 18.2% | 2.0% | 12.0% | **59.6%** |

Table 2: **Quantitative comparisons** on video-to-4D generation. All metrics are averaged on four ground truth novel views and one input view within the first 25 frames.

| | Consistent4D | STAG4D | DG4D | Efficient4D | Ours |
|---|---|---|---|---|---|
| **CLIP score ↑** | 0.905 | 0.920 | 0.885 | 0.917 | **0.922** |
| **LPIPS ↓** | 12.81 | 12.78 | 16.17 | 14.66 | **12.35** |
| **FVD ↓** | 893.67 | 855.84 | 1143.59 | 873.25 | **831.32** |
| **Gen. time ↓** | 120 mins | 90 mins | 11 mins | 12 mins | 12 mins |

smoothness and temporal consistency, while exhibiting fewer floaters and more complete geometry. The fewer artifacts arise from the ability to directly sample the natural image matrix by harnessing the orthogonal diffusion priors. Additionally, the closeness between both stages indicates that our method achieves commendable spatiotemporal consistency, which is also manifested by the cleaner geometry demonstrated in the additional results from supplementary videos and the appendix. Furthermore, the denoising process of the proposed method is highly parallelizable, which provides a significant efficiency advantage over those that need recurrent evaluation of diffusion UNet.

### 4.3 QUANTITATIVE COMPARISONS

We also present quantitative results on two widely explored 4D generation tasks, video-to-4D and image-to-4D. For video-to-4D task, we report the LPIPS and CLIP similarity on the test split following the previous work (Jiang et al., 2024b), which mainly focus on semantic and perceptual consistency. These metrics are measured on the the views at azimuths of 0°, -75°, 15°, 105°, and 195° within the first 25 frames. The results in table 2 indicate that our method is able to more accurately infer geometry from the reference video with a good trade-off between generation quality and time. And the lower FVD suggests that our method achieves better temporal coherence. We also conduct a user study for both video-to-4D and image-to-4D tasks. The survey primarily focused on three aspects of 4D consistency, *i.e.*, the consistency with the reference view (Ref. consistency) and between the different views (Multi-view consistency) or frames (temporal smoothness); additionally, we also requested participants to report on their subjective preference of overall model quality. The results are shown in table 1. It suggests that our method garnered the highest preference in both the consistency dimension and overall model quality. Under the setting described in section 4.1, our method can complete the generation process within 20 minutes, where the first takes 8m31s. It is significantly faster than the state-of-the-art SDS-based methods with sophisticated multi-stage optimization. It is noteworthy that the parallel denoising characteristic could offer an extra trade-off between memory footprint and inference time. The reported times are measured on 8 NVIDIA RTX A6000 GPUs, which have not fully exploited their parallel capabilities. When the number of GPUs is increased to 16, the generation time is expected to be halved. On the other hand, we can also sacrifice some parallelism to achieve a more friendly memory footprint. Further substantial efficiency improvements can be achieved by optimizing the multi-GPU communication, which contributes mostly to the gap between the theoretically optimal inference time.

### 4.4 ABLATION STUDIES

We perform the following ablations to verify the effectiveness of our core designs, and present the results in figure 6. Please refer to the videos in supplementary materials for a more comprehensive and intuitive comparison.

**The impact of variance-reducing sampling** We first analyze the effect of the proposed variance-reducing sampling (VRS). In figure 6, we show the images synthesized in the first stage with and without applying this technique. It can be obviously observed that without the variance-reducing

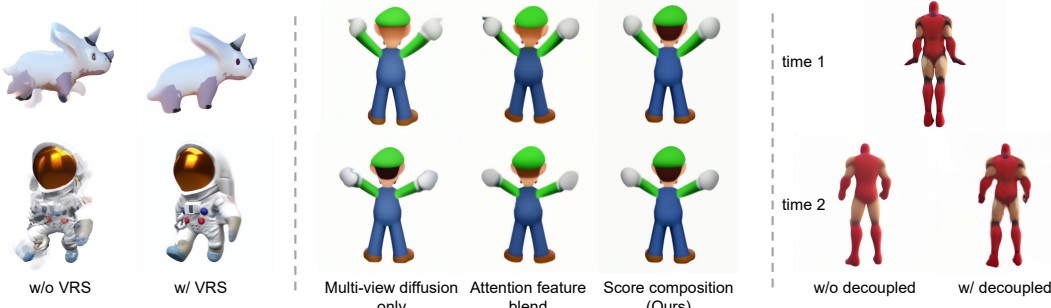

Figure 6: **Ablation studies** for variance-reducing sampling (left), score composition (middle), and texture-dynamic decoupled reconstruction (right).

sampling, the generated images may exhibit severe ghosting artifacts. This arises from the discrepancy between the learned distribution of the two models, which may lead to substantial deviation in the denoising directions, especially in the first few steps. To support this claim, we visualize the predicted $x_0$ of the two models in the 4-th step in figure 5, where two distinct modes can be seen. These discrepancies can be harmonized at higher noise levels via variance-reducing sampling, which is ascribed for the effectiveness of VRS.

**The impact of score composition** The score composition is essential in our framework. To demonstrate its efficacy, we compared the images synthesized in three alternative settings in stage-1: (1) Generating multi-view images independently for each frame using SV3D; (2) Blending the key and value of the spatial attention within SV3D in the time dimension using 1D conv; (3) Using our score composition as equation 5. We enable variance-reducing sampling for all three settings for fair comparison. A couple of observations can be drawn from figure 6:

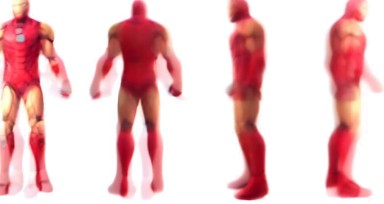

Figure 5: **Visualization of two isolated modes in the early denoising process.**

(1) While variance-reducing sampling can control variance to some extent, significant inconsistency still can be observed at the most under-constraint back views with only multi-view generative model; (2) Due to the limited receptive field and hand-crafted fixed conv weights, blending the attention feature between adjacent frames in each step of denoising is challenging to maintain long-range consistency. Another similar scheme is blending the feature of each frame with the reference frame. However, such a pixel-wise mixing approach also struggles to handle the long-range spatial context, resulting in ghosting effects on large movements. In contrast, the score composition can support arbitrary length of temporal and spatial context, thus can generate seamless videos.

**The impact of decoupled reconstruction** Additionally, we investigate the effect of decoupled reconstruction of texture and geometry in the second stage. In the first stage, we adopted an aggressive setting for variance-reducing sampling to minimize its impact on efficiency, but this led to a gradual blurring of details in images as they moved farther from the reference view. Nevertheless, these details were still relatively well-preserved in the reference frame. Therefore, as depicted in figure 6, our stage-2 was capable of effectively restoring them through decoupled reconstruction even for the view farthest from the references.

## 5 CONCLUSION

In this work, we introduce *Diffusion²*, a novel framework for creating dynamic 3D content. This framework first generates a dense array of multi-view and multi-frame images with high parallelization, which are then used to build a full 4D representation through a reconstruction pipeline. The core assumption behind this framework is that the motion of an object viewed from one angle and its appearance from another are conditionally independent. Based on this assumption, we prove that we can directly sample synchronized multi-view videos in a denoising process by combining pretrained video and multi-view diffusion models. Our experimental results demonstrate the flexibility and effectiveness of our framework, showing that it can adapt to various prompts to produce high-quality 4D content efficiently and effectively. We hope that our work can inspire future research on unleashing the geometrical and dynamic priors from foundation 3D and video diffusion models.

## ACKNOWLEDGMENT

This work was supported in part by National Natural Science Foundation of China (Grant No. 62376060).

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

# A APPENDIX

## A.1 LIMITATIONS

Beyond demonstrating the efficacy of the proposed approach, as a preliminary exploration of photogrammetry-based 4D generation via zero-shot sampling of multi-view synchronized videos within a single pass of denoising diffusion procedure, this paper acknowledges that it still has some limitations.

First, assumption 3.1 may not strictly hold for some cases. However, minor violations do not substantially affect the quality of generated results. As shown in the experimental results, our score composition framework is capable of generating smooth and consistent multi-view videos even for cases not adhering to this assumption. We provide a detailed discussion about its necessity on the correctness of theroem 3.1 in section A.4.

Second, our framework's superiority in efficiency originates from its parallelism. It actually introduces a trade-off between memory and efficiency, which leads to a higher maximum inference efficiency crucial for some online tasks, while not resulting in similar advantages on throughput which is important for some offline tasks. We illustrate this trade-off in figure 7. It can be observed that even using a single GPU, our method still surpasses representative SDS-based alternates in efficiency by a considerable margin.

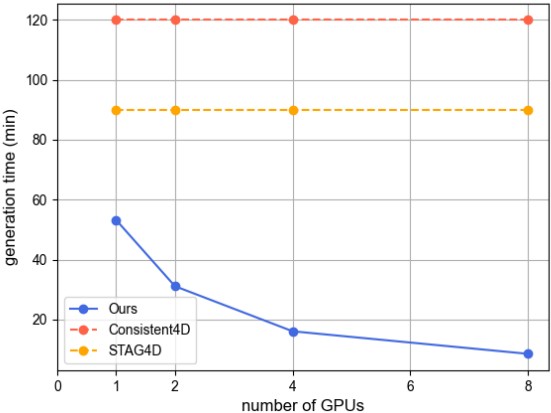

Figure 7: **Inference time of the first stage with different number of GPUs.** For comparison, we also present the inference time of two representative SDS-based solutions. They are shown as horizontal lines, because their efficiency cannot benefit from more computes.

Last, since we adopt a radical VRS setting for efficiency, texture diversity for some views in image matrix far from the references may be sacrificed for some cases. For example, as can be seen from the second example in figure 4 (a), the model has generated a weird texture-less back for the cat. (Actually, for some input images out of training distribution of multi-view prior, the generated multi-view references $\mathcal{I}_{0,\{1:V\}}$ already pose textureless backside view. To more clearly distinguish the extent to which this can be attributed to the proposed sampling strategy, in figure 8, we show the multi-view references of the samples conditioned on generated images in figure 4.)

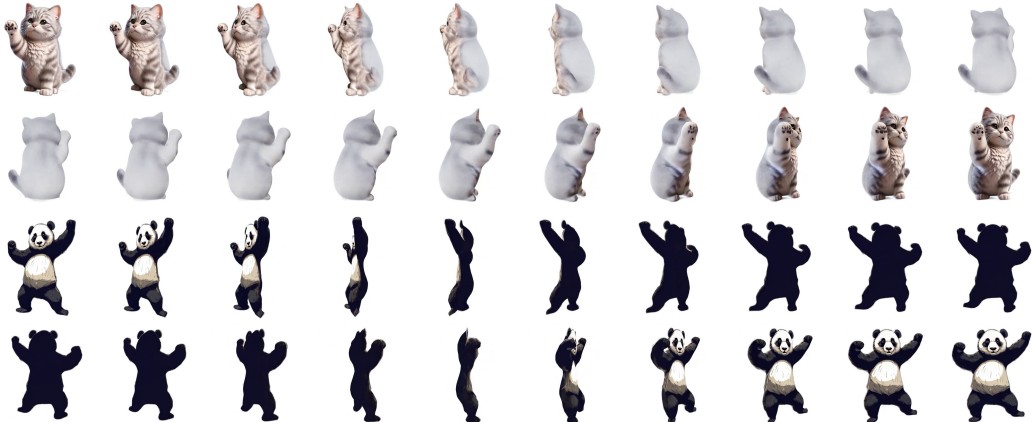

Figure 8: **Multi-view references $\mathcal{I}_{\cdot,\{1:V\}}$ of samples from out-of-distribution images in figure 4.**

Fortunately, due to details are still preserved in the reference frame, some techniques can be employed to transfer them from the reference frame into those texture-less views during the reconstruction process. We adopt a Gaussian-based dynamic-texture decoupling reconstruction pipeline to address this issue. Besides, since this phenomenon does not impact the training of geometry and deformation fields due to its low-frequency nature, further decoupling can be achieved by extracting deformable meshes after their optimization, and independently refining textures using the reference frame. Beyond that, the issue also holds potential for avoidance in the first stage, which we leave for future work.

We sincerely hope that our exploration and results could demonstrate the feasibility and potential of zero-shot sampling of multi-view videos using existing diffusion priors, thereby opening up a brand-new path for 4D generation that is not constrained by the bottleneck of expensive and hard-to-scale 4D data.

### A.2  PSEUDOCODE FOR JOINT DENOISING OF MULTI-VIEW MULTI-FRAME IMAGE MATRIX VIA SCORE COMPOSITION

### A.3  THE PROOF FOR MAIN THEOREM 3.1

Recall that the assumption 3.1 add a conditional independence property on the probability structure of $\mathcal{I}$. However, in the denoising process, the image matrix $\mathcal{I}$ is perturbed with Gaussian noise, so we need a similar property on the noisy version of $\mathcal{I}$, which leads to corollary A.1:

**Corollary A.1.** *Denote $\hat{\mathcal{I}}$ as the noisy version of $\mathcal{I}$, i.e.,*

$$\hat{\mathcal{I}} = \{\hat{I}_{i,j} \in \mathbb{R}^{H \times W \times 3}\}_{i=1,j=1}^{V,F} \quad \text{with } \hat{I}_{i,j} = \alpha I_{i,j} + \varepsilon_{i,j}, \tag{7}$$

*where $\alpha \in \mathbb{R}$ is a constant and $\varepsilon_{i,j} \in \mathbb{R}^{H \times W \times 3}$ are independent Gaussian noises. Then equation 4 also holds for $\hat{\mathcal{I}}$:*

$$p\left(\hat{\mathcal{I}}_{-i,j}, \hat{\mathcal{I}}_{i,-j} | \hat{I}_{i,j}\right) = p\left(\hat{\mathcal{I}}_{-i,j} | \hat{I}_{i,j}\right) p\left(\hat{\mathcal{I}}_{i,-j} | \hat{I}_{i,j}\right). \tag{8}$$

Then we can set out proving the main theorem 3.1.

---

**Algorithm 1** Joint denoising for multi-view multi-frame image matrix via score composition.

---

**Require:** Initial noise $\mathcal{I}^0$, multi-view denoiser $D_\theta(\boldsymbol{x};\sigma)$, video denoiser $D_\phi(\boldsymbol{x};\sigma)$, discrete noise levels $\{\sigma^j\}_{j\in[1\cdots N]}$, scale schedule $s$, rollback steps $N_r$

1: **for all** $i \in \{0, \cdots, N-1\}$ **do**
2:    $R \leftarrow 2$ **if** $i \in \{0, \cdots, N_r - 1\}$ **else** 1
3:    **for all** $j \in \{0, \cdots, R-1\}$ **do**
4:      $\boldsymbol{d}^i \leftarrow \text{zero\_like}(\mathcal{I}^i)$
5:      **for all** $v \in \{0, \cdots, V-1\}$ **do**
6:        $\boldsymbol{d}^i_{\{v,:\}} \leftarrow \boldsymbol{d}^i_{\{v,:\}} + (1 - s(i))(\mathcal{I}^i_{\{v,:\}} - D_\phi(\mathcal{I}^i_{\{v,:\}}; \sigma_i))/\sigma_i$
7:      **end for**
8:      **for all** $f \in \{0, \cdots, F-1\}$ **do**
9:        $\boldsymbol{d}^i_{\{:,f\}} \leftarrow \boldsymbol{d}^i_{\{:,f\}} + s(i)(\mathcal{I}^i_{\{:,f\}} - D_\theta(\mathcal{I}^i_{\{:,f\}}; \sigma_i))/\sigma_i$
10:      **end for**
11:      $\mathcal{I}^{i+1} \leftarrow \mathcal{I}^i + (\sigma_{i+1} - \sigma_i)\boldsymbol{d}_i$
12:      **if** $j < R-1$ **then**
13:        **sample** $\epsilon \sim \mathcal{N}(\mathbf{0}, \mathbf{I})$
14:        $\mathcal{I}^i \leftarrow \mathcal{I}^{i+1} + (\sigma_i - \sigma_{i+1})\epsilon$
15:      **end if**
16:    **end for**
17: **end for**
18: **return** $\mathcal{I}^N$

---

*Proof.* We first decompose $p\left(\hat{\mathcal{I}}\right)$ by

$$p\left(\hat{\mathcal{I}}\right) = p\left(\hat{I}_{i,j}, \hat{\mathcal{I}}_{-i,j}, \hat{\mathcal{I}}_{i,-j}, \hat{\mathcal{I}}_{-i,-j}\right) = p\left(\hat{I}_{i,j}, \hat{\mathcal{I}}_{-i,-j} | \hat{\mathcal{I}}_{-i,j}, \hat{\mathcal{I}}_{i,-j}\right) p\left(\hat{\mathcal{I}}_{-i,j}, \hat{\mathcal{I}}_{i,-j}\right). \quad (9)$$

Note that $\forall \hat{I}_{i',j'} \in \hat{\mathcal{I}}_{-i,-j}$, $I_{i',j'}$ and $I_{i,j}$ are independent conditioned on $I_{i',j}$ by corollary A.1, hence

$$p\left(\hat{I}_{i,j}, \hat{\mathcal{I}}_{-i,-j} | \hat{\mathcal{I}}_{-i,j}, \hat{\mathcal{I}}_{i,-j}\right) = p\left(\hat{\mathcal{I}}_{-i,-j} | \hat{\mathcal{I}}_{-i,j}, \hat{\mathcal{I}}_{i,-j}\right) p\left(\hat{I}_{i,j} | \hat{\mathcal{I}}_{-i,j}, \hat{\mathcal{I}}_{i,-j}\right). \quad (10)$$

Since $p\left(\hat{\mathcal{I}}_{-i,-j} | \hat{\mathcal{I}}_{-i,j}, \hat{\mathcal{I}}_{i,-j}\right)$ does not contain x ($=I_{i,j}$), its derivative with respect to x is zero thus this term does not contribute to the score of $I_{i,j}$. Then combined with equation 9 and equation 10, taking the derivative of $\log p\left(\hat{\mathcal{I}}\right)$ with respect to x, we achieve

$$\begin{aligned} \nabla_{\mathrm{x}} \log p\left(\hat{\mathcal{I}}\right) &= \nabla_{\mathrm{x}} \log p\left(\hat{I}_{i,j} | \hat{\mathcal{I}}_{-i,j}, \hat{\mathcal{I}}_{i,-j}\right) p\left(\hat{\mathcal{I}}_{-i,j}, \hat{\mathcal{I}}_{i,-j}\right) \\ &= \nabla_{\mathrm{x}} \log p\left(\hat{I}_{i,j}, \hat{\mathcal{I}}_{-i,j}, \hat{\mathcal{I}}_{i,-j}\right). \end{aligned} \quad (11)$$

Finally, by further decomposing $p\left(\hat{I}_{i,j}, \hat{\mathcal{I}}_{-i,j}, \hat{\mathcal{I}}_{i,-j}\right)$ and directly applying corollary A.1, we obtain

$$\begin{aligned} \nabla_{\mathrm{x}} \log p\left(\hat{\mathcal{I}}\right) &= \nabla_{\mathrm{x}} \log p\left(\hat{\mathcal{I}}_{-i,j}, \hat{\mathcal{I}}_{i,-j} | \hat{I}_{i,j}\right) p\left(\hat{I}_{i,j}\right) \\ &= \nabla_{\mathrm{x}} \log p\left(\hat{\mathcal{I}}_{-i,j} | \hat{I}_{i,j}\right) p\left(\hat{\mathcal{I}}_{i,-j} | \hat{I}_{i,j}\right) p\left(\hat{I}_{i,j}\right) \\ &= \nabla_{\mathrm{x}} \log \frac{p\left(\hat{\mathcal{I}}_{\{1:V\},j}\right) p\left(\hat{\mathcal{I}}_{i,\{1:F\}}\right)}{p\left(\hat{I}_{i,j}\right)} \\ &= \nabla_{\mathrm{x}} \log p\left(\hat{\mathcal{I}}_{\{1:V\},j}\right) + \nabla_{\mathrm{x}} \log p\left(\hat{\mathcal{I}}_{i,\{1:F\}}\right) - \nabla_{\mathrm{x}} \log p\left(\hat{I}_{i,j}\right). \end{aligned} \quad (12)$$

$\square$

### A.4 THE APPLICABILITY OF THE ASSUMPTION 3.1 AND PROOF FOR THE THEOREM 3.1 IN GENERAL CASE

Another risk that needs to be pointed out is that our assumption about the conditional independence is too strong to apply in some cases. For example, if the target object rotates 180° over a period of time, the back view at the current moment and the front view after a period of time should not be conditionally independent given the front view at the current moment. Fortunately, existing video diffusion models typically generate videos from a single view without rotational movements, and our pipeline allows users to specify conditions to exclude such cases.

In addition, it is worth noting that even in such cases that violate assumption 3.1, our method can still work well. This is because the main theorem remains valid under these case and its correctness does not essentially rely on such a strong assumption. Although extreme rotations might exist when considering frames with large intervals, they are unlikely to occur between adjacent frames in a natural video. Now considering that the distribution of $\hat{I}_{i,j}$ is almost entirely determined by its adjacent frames and views, we have the continuity assumption:

**Assumption A.1.** *(**Continuity assumption**) Denote $\hat{\mathcal{I}}_{i_{near},j} \in \hat{\mathcal{I}}_{-i,j}, \hat{\mathcal{I}}_{i,j_{near}} \in \hat{\mathcal{I}}_{i,-j}$ are the views and frames close enough to $\hat{I}_{i,j}$ to provide sufficient information for the distribution of it while keeping sufficiently continuous, then*

$$p\left(\hat{I}_{i,j}|\hat{\mathcal{I}}_{-i,j}\right) \approx p\left(\hat{I}_{i,j}|\hat{\mathcal{I}}_{i_{near},j}\right) \tag{13}$$

$$p\left(\hat{I}_{i,j}|\hat{\mathcal{I}}_{i,-j}\right) \approx p\left(\hat{I}_{i,j}|\hat{\mathcal{I}}_{i,j_{near}}\right) \tag{14}$$

$$p\left(\hat{I}_{i,j}|\hat{\mathcal{I}}_{i_{near},j}, \hat{\mathcal{I}}_{i,j_{near}}, \hat{I}_{i',j'}\right) \approx p\left(\hat{I}_{i,j}|\hat{\mathcal{I}}_{i_{near},j}, \hat{\mathcal{I}}_{i,j_{near}}\right), \tag{15}$$

*where $\hat{I}_{i',j'} \notin \{\hat{I}_{i,j}, \hat{\mathcal{I}}_{i_{near},j}, \hat{\mathcal{I}}_{i,j_{near}}\}$.*

Further note that the near index is symmetric, *i.e.*, if $\hat{I}_{i',j} \in \hat{I}_{i_{near},j}$, then $\hat{I}_{i,j} \in \hat{I}_{i'_{near},j}$. Then we relax the assumption 3.1 as follows:

**Assumption A.2.** *(**Weak version of assumption 3.1**) Given any image $I_{i,j}$, the nearby geometry $I_{i',j} \in \mathcal{I}_{i_{near},j}$ and the nearby dynamics $I_{i,j'} \in \mathcal{I}_{i,j_{near}}$ are conditionally independent, i.e.,*

$$p\left(I_{i',j}, I_{i,j'}|I_{i,j}\right) = p\left(I_{i',j}|I_{i,j}\right) p\left(I_{i,j'}|I_{i,j}\right). \tag{16}$$

*Similar condition also holds for $\hat{\mathcal{I}}$.*

Finally, we can establish our main theorem 3.1 under the additional assumptions.

*Proof of theorem 3.1 under assumption A.1 and assumption A.2.*

Following the same derivation of the original proof A.3, equations (9), (10) and (11) remain unchanged but all "=" in (10) and (11) should be "≈" because equation 10 is now established on equation 15 of assumption A.1 instead of assumption 3.1.

Then integrating new assumptions to correct equation 12, we obtain:

$$
\begin{aligned}
\nabla_{\mathrm{x}} \log p\left(\hat{\mathcal{I}}\right) &\approx \nabla_{\mathrm{x}} \log p\left(\hat{I}_{i,j}|\hat{\mathcal{I}}_{-i,j}, \hat{\mathcal{I}}_{i,-j}\right) p\left(\hat{\mathcal{I}}_{-i,j}, \hat{\mathcal{I}}_{i,-j}\right) \\
&= \nabla_{\mathrm{x}} \log p\left(\hat{I}_{i,j}|\hat{\mathcal{I}}_{-i,j}, \hat{\mathcal{I}}_{i,-j}\right) + \underbrace{\nabla_{\mathrm{x}} \log p\left(\hat{\mathcal{I}}_{-i,j}, \hat{\mathcal{I}}_{i,-j}\right)}_{=0} \\
&\approx \nabla_{\mathrm{x}} \log p\left(\hat{I}_{i,j}|\hat{\mathcal{I}}_{i_{near},j}, \hat{\mathcal{I}}_{i,j_{near}}\right) + \underbrace{\nabla_{\mathrm{x}} \log p\left(\hat{\mathcal{I}}_{i_{near},j}, \hat{\mathcal{I}}_{i,i_{near}}\right)}_{=0} \\
&= \nabla_{\mathrm{x}} \log p\left(\hat{I}_{i,j}|\hat{\mathcal{I}}_{i_{near},j}, \hat{\mathcal{I}}_{i,j_{near}}\right) p\left(\hat{\mathcal{I}}_{i_{near},j}, \hat{\mathcal{I}}_{i,i_{near}}\right) \\
&= \nabla_{\mathrm{x}} \log p\left(\hat{I}_{i,j}|\hat{\mathcal{I}}_{i_{near},j}, \hat{\mathcal{I}}_{i,j_{near}}\right).
\end{aligned}
\tag{17}
$$

This suggests the score of $I_{i,j}$ is only related to $\hat{\mathcal{I}}_{i_{near},j}$ and $\hat{\mathcal{I}}_{i,j_{near}}$. Similar to the derivation in equation 12, we can achieve:

$$\nabla_{\mathrm{x}} \log p\left(\hat{\mathcal{I}}\right) = \nabla_{\mathrm{x}} \log \frac{p\left(\hat{I}_{i,j}, \hat{\mathcal{I}}_{i_{\mathrm{near}},j}\right) p\left(\hat{I}_{i,j}, \hat{\mathcal{I}}_{i,j_{\mathrm{near}}}\right)}{p\left(\hat{I}_{i,j}\right)}. \tag{18}$$

Finally, by further decomposing the numerator and applying the condition of continuity, we have:

$$
\begin{aligned}
\nabla_{\mathrm{x}} \log p\left(\hat{\mathcal{I}}\right) &= \nabla_{\mathrm{x}} \log \frac{p\left(\hat{\mathcal{I}}_{i,j} | \hat{\mathcal{I}}_{i_{\mathrm{near}},j}\right) p\left(\hat{\mathcal{I}}_{i_{\mathrm{near}},j}\right) p\left(\hat{\mathcal{I}}_{i,j} | \hat{\mathcal{I}}_{i,j_{\mathrm{near}}}\right) p\left(\hat{\mathcal{I}}_{i,j_{\mathrm{near}}}\right)}{p\left(\hat{I}_{i,j}\right)} \\
&\approx \nabla_{\mathrm{x}} \log \frac{p\left(\hat{\mathcal{I}}_{i,j} | \hat{\mathcal{I}}_{-i,j}\right) p\left(\hat{\mathcal{I}}_{-i,j}\right) p\left(\hat{\mathcal{I}}_{i,j} | \hat{\mathcal{I}}_{i,-j}\right) p\left(\hat{\mathcal{I}}_{i,-j}\right)}{p\left(\hat{I}_{i,j}\right)} \\
&= \nabla_{\mathrm{x}} \log \frac{p\left(\hat{\mathcal{I}}_{\{1:V\},j}\right) p\left(\hat{\mathcal{I}}_{i,\{1:F\}}\right)}{p\left(\hat{I}_{i,j}\right)} \\
&= \nabla_{\mathrm{x}} \log p\left(\hat{\mathcal{I}}_{\{1:V\},j}\right) + \nabla_{\mathrm{x}} \log p\left(\hat{\mathcal{I}}_{i,\{1:F\}}\right) - \nabla_{\mathrm{x}} \log p\left(\hat{I}_{i,j}\right).
\end{aligned}
\tag{19}
$$

Now we have verified the correctness of the conclusion in equation 12. Based on the similar derivation, we can also ensure the safety of equation 10. This guarantees the correctness of our main theorem. $\square$

## A.5 ADDITIONAL EXPERIMENTAL DETAILS

**Dataset and evaluate setting** In the main text, we test the proposed approach on two types of conditions: reference image and single-view video. The input images and videos used in the qualitative comparisons are sourced from the previous works Zhao et al. (2023); Jiang et al. (2024b); Tang et al. (2024b). For the quantitative evaluation in the video-to-4D task, we evaluate all methods on the Consistent4D test split and report the CLIP similarity, LPIPS, and FVD averaged on five views with accessible ground truth within the first 25 frames. Note that in equation 5, we introduced a coefficient $s$ to modulate the contribution of both models within the convex combination.

**Scale schedule** Note that in equation 5, we introduced a coefficient $s$ to modulate the contribution of both models within the convex combination. Its schedule in denoising process can provide additional design space for enhancing the details at specific views. However, to ease the reproduction and reduce the reliance on manual priors, we fix it at 0.5 in all steps which is sufficient to achieve satisfactory results for most cases.

**User study** Figure 9 shows the template we used for user study. The participants are asked to pick one generation results from five candidates (displayed in video format) according to four dimensions: consistency with the reference video, multi-view consistency, temporal smoothness and overall model quality. Total 50 participants from diverse backgrounds are asked to do the questionnaire. And there are 11 cases for user study, including *anya, cat-wave, patrick-star, flying-ironman, luigi, panda-dance, running-triceratops, sighing-frog, tiger-guitar, trump, walking-astronaut*. All of them are collected from the previous works (Jiang et al., 2024b; Zhao et al., 2023; Tang et al., 2024b).

## A.6 ADDITIONAL RESULTS

### A.6.1 QUANTITATIVE ABLATIONS

In section 4.4 and the supplementary video, we have validated the effectiveness of the core components through qualitative ablations, where the improvements over naïve baselines are clearly and intuitively presented from the provided samples. To provide a more comprehensive evaluation, we present quantitative ablations in table 3, where all metrics are averaged across all seven samples in the Consistent4D (Jiang et al., 2024b) test dataset. We also report the result of using only the multi-view generator. While this approach leads in image-based metrics (*e.g.*, LPIPS, CLIP), due to

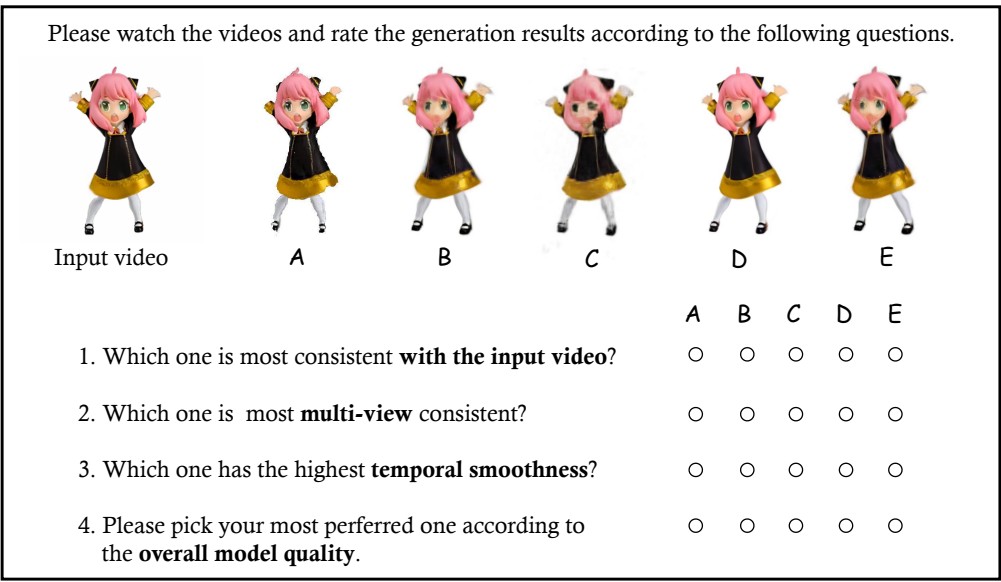

Figure 9: **The layout of our user study.**.

Table 3: **Quantitative ablations.** Each method is benchmarked in the same way as table 2. Disabling score composition means multi-view diffusion only.

|  | VRS | Score composition | Attention feature blend | LPIPS ↓ | CLIP score ↑ | FVD ↓ |
|---|---|---|---|---|---|---|
| (a) | | mv-generation only | | 11.63 | 0.948 | 1248.49 |
| (b) | | (a) + recon with Li et al. (2024) | | 14.86 | 0.896 | 1412.00 |
| (c) | ✓ | | | 12.79 | 0.919 | 1050.68 |
| (d) | ✓ | | ✓ | 12.65 | 0.922 | 898.66 |
| (e) | | | ✓ | 11.86 | 0.943 | 1170.96 |
| (f) | ✓ | ✓ | | 12.35 | 0.922 | 831.32 |
| (g) | | (f) w/o decoupled recon. | | 12.57 | 0.921 | 833.37 |

the inherent stochasticity in the diffusion denoising process, its output exhibits significant temporal inconsistencies, making it challenging to be reconstructed into meaningful 4D assets, even for the state-of-the-art 4D reconstruction method (Li et al., 2024) with superior fitting capability.

### A.6.2 MORE VISUALIZATIONS

In figure 10, we provided more cases for qualitative comparison. Similar conclusion to section 4.2 can be achieved that our results outperform the others. Figures 11 and 12 further show additional results on custom data, with RGB and depth images from two views and three timestamps displayed. We recommend readers to watch the videos in our supplementary materials.

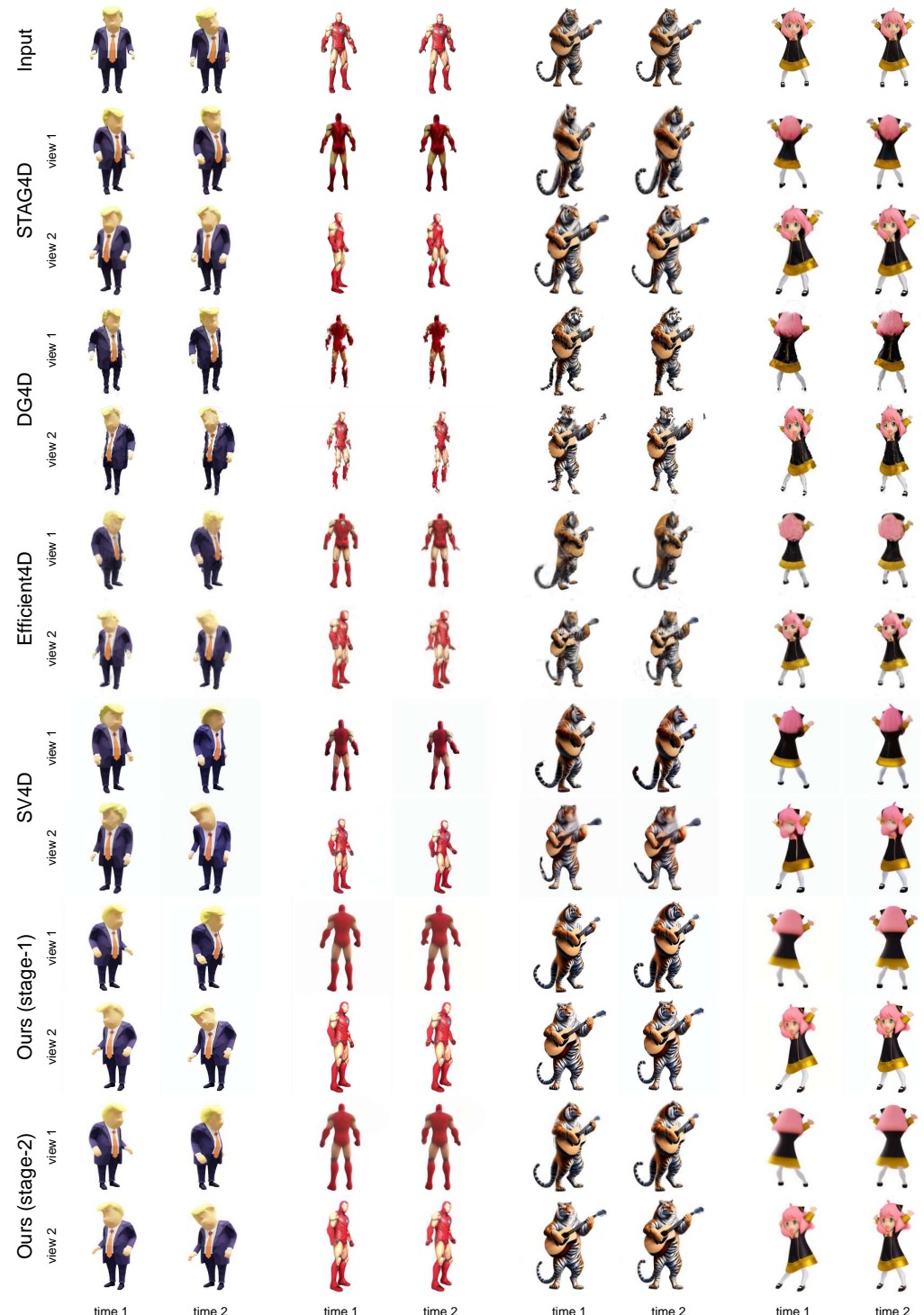

Figure 10: **Qualitative comparisons**. For each method, we show images from two views and two timestamps.

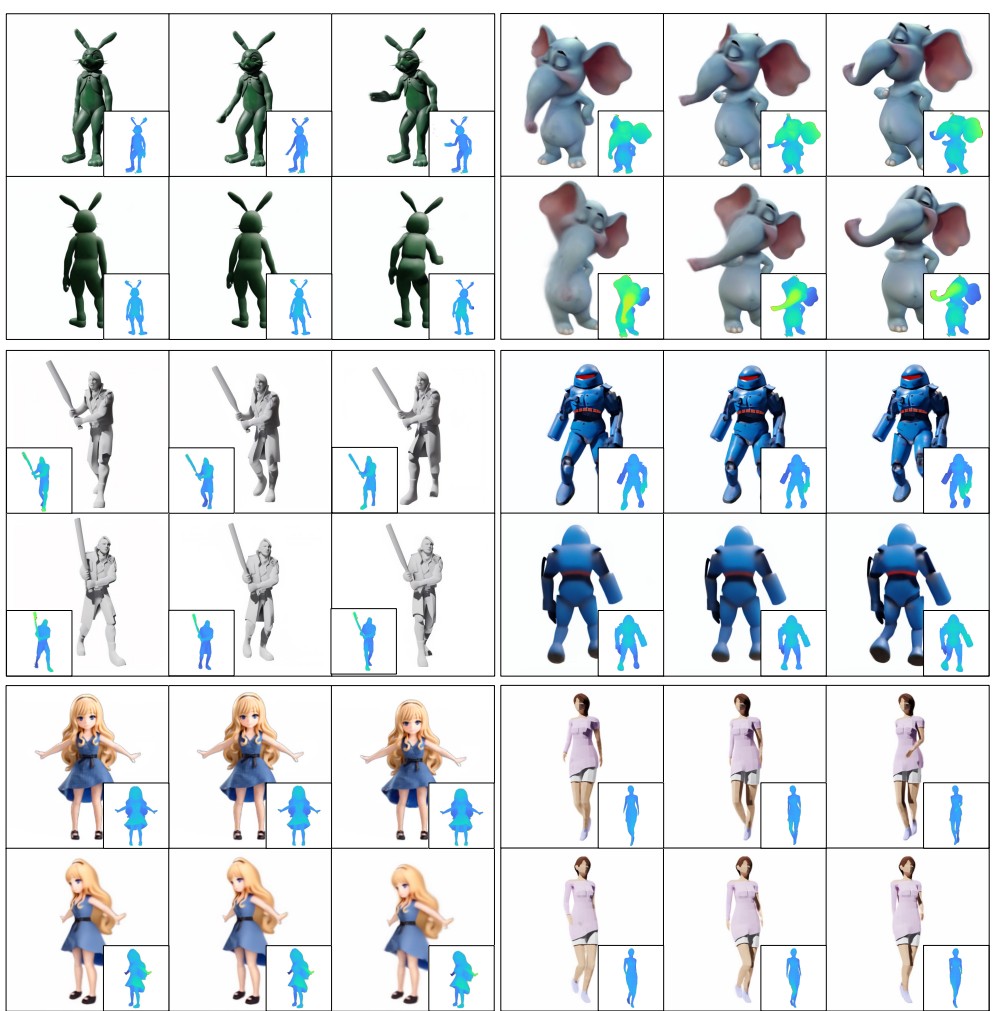

Figure 11: **More visualization results.**

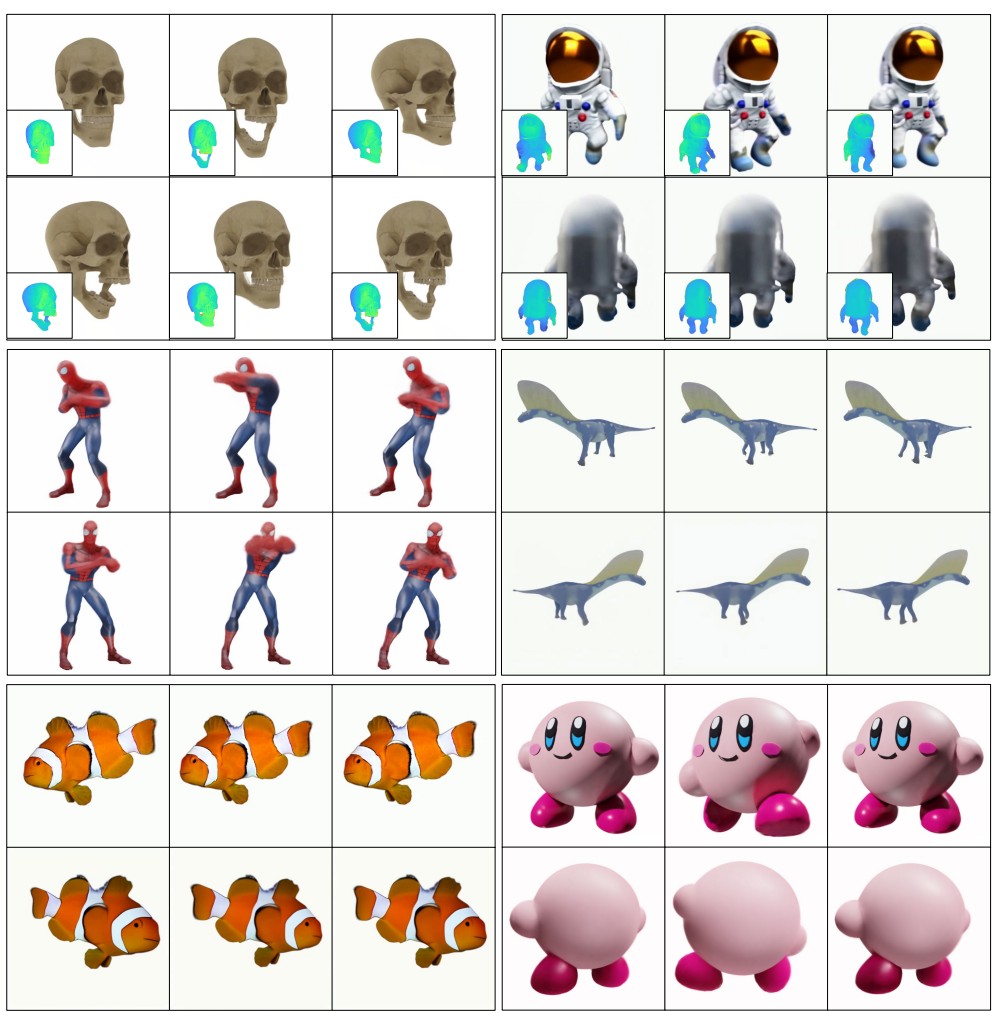

Figure 12: **More visualization results.**

