# OpenReview forum: "Diffusion$^2$: Dynamic 3D Content Generation via Score Composition of Video and Multi-view Diffusion Models"
_ICLR.cc/2025/Conference — ICLR 2025 Poster_

### Official Review · Reviewer_UTNE · 2024-10-17

**Soundness:** 3
**Presentation:** 3
**Contribution:** 3
**Rating:** 8
**Confidence:** 4

**Summary:**

This proposal design a framework for combination between temporally aligned VDM(Video Diffusion Model) and spatially aligned multi-view generation model. It utilizes runtime optimization instead of training to build up 4D objects. Among recent SoTA 4D generation methods, this proposal is free of training and achieves competitive performance. Also, it proves that we can use off-the-shell models for 4D content generation without further training.

**Strengths:**

1、The framework setting needs no further training.
2、The authors write clear proofs for the denoise process and 4D reconstruction process.
3、Experiments are sufficient to prove the anticipated conclusion.

**Weaknesses:**

Some conclusions are not that rigorous, see questions.

**Questions:**

1、Figure3，If the viewpoints differ by only 30 degrees, can they still be considered independent variables?
2、Variance-reducing sampling，the rollback operation need more specific explanation of why it works? And demonstrate why this would reduce the variance of sampling results? Also, If we set the sampling steps larger, will it achieves similar effect with Variance-reducing sampling?

---

> ### Author Response · Authors · 2024-11-22
> **Response to Reviewer UTNE**
>
> We thank the reviewer for the positive feedback and insightful comments. Our responses to the reviewer’s concerns are below:
>
> **Q1: If the viewpoints differ by only 30 degrees, can they still be considered independent variables?**
>
> Yes, although these two views may have some overlapping regions, the **conditional** independence still holds. Because the case in Figure 3 does not contain some special motions like twisting. The overlap between the future front view $I_{i,j’}$ and the current side view $I_{i’,j}$ also exists in the current front view $I_{i,j}$. In practice, even when twisting or rotation movement occurs, it is often subtle between frames, contributing negligible information gain and thus having no substantial violation on the conditional independence assumption. As demonstrated at **in the supplementary video (01:04~01:08)**, our method is also capable of handling objects with such dynamics.
>
> **Q2: More explanation for why VRS works and why it would reduce the variance.**
>
> Thanks.
> We have explained its motivation in Section 3.3 and provide further clarification here.
>
> The similar rollback operations have been adopted in many diffusion-based inpainting methods [1] for harmonizing potential conflicts.
>
> In our work, the utilization of this operation aims to deceive the diffusion model into denoising a noisy latent with a smaller variance at a higher noise level through the following process: before $t$-th denoising step, we add Gaussian noise with a variance smaller than $\sigma_{t+1}^2-\sigma_{t}^2$ onto the latent, and then denoise it again at noise level $t+1$.
>
> After that, at $t$-th step, the noisy latent to be denoised by the model actually resides at a level below $\sigma_t$.
>
> We provide a visualization of how variance evolves during denoising with and without VRS **in Figure 10** of the revised appendix.
>
> A more intuitive explanation for why it works is that when the latent resides at a lower noise level, the diffusion model can allocate more capacity to reconcile the conflicts brought by two scores. Additionally, a latent with less noise can provide clearer guidance for denoising, which can also help to mitigate potential conflicts.
>
> **Q3: If we set the sampling steps larger, will it achieve a similar effect with variance-reducing sampling?**
>
> We would say no.
>
> In the **revised supplementary video (00:12~00:21)**, we provide two videos generated using 500 denoising steps (10x than the default setting) without VRS.
>
> These assets still exhibit severe flickering artifacts.
>
> This may because the unresolved conflicts in the early stage of denoising will drive the latent away from the desired distribution, which cannot be rectified by merely extending the denoising schedule.
>
> > [1] Lugmayr, Andreas, et al. Repaint: Inpainting using denoising diffusion probabilistic models. *CVPR*. 2022.

---

> ### Comment · Reviewer_UTNE · 2024-11-23
>
> Thanks for your clarification. With the development of 3D generation and Video generation, they both converge to some standard formulation. Like transformer-based regression 3D-Gen method(Clay, LRM..)  and CasualVAE+DiT-based Video Generation method(OpenSoraPlan, CogVideo-X). However, 4D generation is still under development. Though I think composition of both SoTA methods is far from perfect, I would like to see more researchers make their works open-source. I will keep my rating until I see final comments from other reviewers.

---

> > ### Author Response · Authors · 2024-11-23
> >
> > Thanks again for reviewer's invaluable time and efforts dedicated to reviewing our paper and providing insightful comment which demonstrates a deep understanding and knowledge of the latest development in this field.
> >
> > Actually, the 3D prior adopted in this work is not a native 3D generative model like CLAY or LRM, but rather a diffusion-based multi-view image generative model.
> > Compared to the native 3D generative models which often rely on large-scale and diverse 3D datasets, the multi-view image generator retains its unique advantages as it has the potential to break the ceilings brought by the scale of available 3D data and extend to more diverse scenarios.
> > Specifically, a multi-view image generative model intertwined with a video diffusion model in the same latent space to model their marginal distributions, and sample for the purpose of a continuous 4D assets.
> > We hope this new solution could pave a way for the community, that can circumvent the need to scale expensive 4D data.
> >
> > However, the reviewer's suggestion of integrating native 3D generative models with the SoTA video generative models for 4D generation is an interesting direction.
> > We would like to continue exploring better integration of them in the future.
> >
> > The authors also notice that some works in this area is not open-sourced. However, we are willing to release all code and models of this work to foster the development and transparency in the field of 4D generation.
> >
> > Hope our responses clarify above thoughtful questions, and it is very much appreciated if the reviewer can kindly check our responses and provide feedback with further questions/concerns (if any). We would be more than happy to address them. Thank you!

---

> ### Comment · Reviewer_UTNE · 2024-11-26
>
> Sincerely speaking, I should give a weak accept(6) due to the novelty at present time. But actually, many follow-up works like Vidu4D, SV4D are inspired by this formulation(marrying VSD and Multi-View Denoiser). So I strongly recommend accepting this work as a milestone of  4D generation. (The quality gap lies in the ability of foundation video diffusion model but not the framework.)

---

> > ### Author Response · Authors · 2024-11-26
> >
> > We sincerely thank reviewer UTNE for acknowledging the novelty and contribution of our work, and really hope our solution could pave a new way for the 4D generation community!

---

### Official Review · Reviewer_rKU7 · 2024-10-29

**Soundness:** 3
**Presentation:** 3
**Contribution:** 3
**Rating:** 6
**Confidence:** 4

**Summary:**

This paper introduces a new algorithm to generate 4D content from a single image or video of a foreground object.
The key idea is to combine a video diffusion model with a multiview diffusion model to generate a set of multiview-consistent videos.
The generation process involves interleaving the denoising of the video diffusion model and the denoising of the multiview diffusion model. To further improve the generation quality, a variance reduction method is proposed to re-denoise some views. Finally, a dynamic Gaussian field is reconstructed from the generated multiview videos.
Experiments demonstrate that the proposed method is able to produce multiview consistent videos and generate 4D contents.

**Strengths:**

The main strength of the paper is that the idea of combining two diffusion models to generate 4D content is novel and interesting.
Due to the lack of 4D data, it is difficult to train a 4D diffusion model directly. This method only takes advantage of video diffusion models and multiview diffusion models, which can be trained separately on large-scale 3D object datasets or video datasets. Thus, this method is promising to avoid directly using 4D data but still allow 4D generation.

The theoretical analysis of the proposed method is reasonable and convincing. The assumptions are provided for a better understanding of the mechanism of the interleaved denoising of two diffusion models.

**Weaknesses:**

The main weakness is the quality of the 4D generation.
The generation of the proposed method seems to be oversmoothed on unseen viewpoints. As we can see in both figures and the supplementary video, the backside all seems to be oversmoothed without any details, especially when the variance reduction is applied.
I guess this may be due to the difficulty in making two diffusion models agree with each other. The assumption of independent probability distribution may be a little bit strong.
The results are even worse than the models with SDS losses or some other models directly trained on limited 4D data.

**Questions:**

I'm not sure whether a finetuning of SVD on videos containing only foreground objects would be helpful in improving the quality. In my experience, SVD performs poorly in generating videos of only one dynamic foreground object.

---

> ### Author Response · Authors · 2024-11-22
> **Response to Reviewer rKU7**
>
> We thank the reviewer for the positive feedback and insightful comments. Our responses to the reviewer’s concerns are below:
>
> **Q1: The backside seems to be over-smoothed.**
>
> Thanks.
>
> To reconcile the scores from the two generative models without significantly impacting runtime, we adopted the relatively aggressive variance-reducing sampling (VRS) strategy, which sacrifices details of some cases in specific views that are under-constraint due to far from the reference frame and view. Our rationale for this trade-off is based on the following considerations:
>
> (i) This compromise does not affect the consistency of the generated results, it is still a seamless 4D assets that is consistent with the reference view.
>
> (ii) It does not degrade the details in views closer to the reference frame and view. Actually, as demonstrated in the video, our results often exhibit better detail in these views compared to other alternates thanks to the consistent direct photometric supervision in high resolution. In some applications, users may more care about quality in these views rather than back views without any hint.
>
> (iii) Finally, this issue can be mitigated by texture-dynamic decoupled reconstruction as shown in the ablations in Figure 6 of the main paper and 00:03~00:11 of the supplementary video.
>
> Besides, this issue could be partly ascribed to the multi-view prior. We observed that for some examples, their multi-view references already lack details in backside as shown **in Figure 8**. Further scaling of the used diffusion prior may alleviate this issue.
>
> **Q2: Whether finetuning the SVD on the object-centric data would be helpful.**
>
> Thanks for the insightful comment.
>
> The authors agree with this point, that fine-tuning the SVD on videos containing only foreground objects could improve quality.
>
> The current adopted video generator has learned a broader distribution than the desired single-view videos for dynamic objects, which may introduce some conflicts.
>
> We also raise this concern and propose VRS to mitigate potential conflicts.
>
> However, to preserve the training-free merits of the proposed framework, we have not included further fine-tuning of the video generative model as part of our contribution. We would like to attempt this direction in the future.

---

> > ### Comment · Reviewer_rKU7 · 2024-11-23
> >
> > Thanks for the responses.
> > 1. I still have some concerns about baseline SDS-based methods like SC-4D or DreamGaussian4D that often produce less smoothed textures on unseen views than the proposed method.
> > 2. Multiview diffusion models like like Wonder3D, and Zero123++,  are able to produce high-quality images with sharp textures so I think this problem is not caused by the MVDiffusion methods.
> > 3. I think training-free is not a key feature here. You can train on 3D models or 2D videos instead of training on multiview videos. The model does not work well without any fine-tuning.
> >
> > In summary, I think the authors propose an interesting idea but the implementation does not fully show the potential of the idea of interleaving denoising of video diffusion and multiview diffusion. The results are reasonable but not too impressive. I fully understand that retraining may require lots of GPU resources and I would keep my positive score here.

---

> > > ### Author Response · Authors · 2024-11-23
> > >
> > > Thanks again for reviewer's invaluable time and efforts to review our work as well as the insightful comment.
> > >
> > > > 1. I still have some concerns about baseline SDS-based methods like SC-4D or DreamGaussian4D that often produce less smoothed textures on unseen views than the proposed method.
> > >
> > > We have included comparisons with STAG4D and DreamGaussian4D in the supplementary video, where our result demonstrates obviously overall better results than them.
> > > Specifically, our results exhibit better detail and fewer artifacts in the visible view. Even for the unseen backside views, our results still maintain fewer artifacts in the examples presented in the video. This is because the **direct photometric supervision** with high consistency can provide more stable gradient in the optimization of Gaussians than SDS-based iterative refinement.
> > >
> > > > 2. Multiview diffusion models like like Wonder3D, and Zero123++, are able to produce high-quality images with sharp textures so I think this problem is not caused by the MVDiffusion methods.
> > >
> > > Thanks. As mentioned in our previous response, we only **partially** ascribe this issue to MVDiffusion.
> > > We provide two examples in Figure 8 to support our speculation.
> > > Specifically, they are multi-view images directly generated by the adopted MVDiffusion conditioned on two out-of-distribution images.
> > > It can be seen that they already exhibit textureless back views.
> > > We also agree with the reviewer that this issue is not found in the results of Zero123++ and Wonder3D, and the issue observed in the SV3D could stem from the unique characteristics of video-diffusion-based multi-view image generators.
> > >
> > > > 3. I think training-free is not a key feature here. You can train on 3D models or 2D videos instead of training on multiview videos. The model does not work well without any fine-tuning.
> > >
> > > The "training-free" property can be viewed as "plug-in" or "architecture-agnostic".
> > > Our score composition approach aims to show the feasibility of applying any two multi-view and video diffusion models together, providing a new solution for the 4D generation community, which can circumvent the need to scale expensive 4D data.
> > > The authors also agree that a better video diffusion model finetuned on 3D datasets will offer further improvement, which we will explore in the future.
> > >
> > > Overall, the questions raised by the reviewer point a promising improvement direction for our work. Hope our responses clarify above thoughtful questions, and it is very much appreciated if the reviewer can kindly check our responses and provide feedback with further questions/concerns (if any). We would be more than happy to address them. Thank you!

---

### Official Review · Reviewer_AvRx · 2024-11-02

**Soundness:** 2
**Presentation:** 3
**Contribution:** 2
**Rating:** 6
**Confidence:** 3

**Summary:**

This paper aims to leverage existing multi-view 3D generation models and video generation models for 4D generation. For this, it designs a score composition strategy to make use of these two kinds of generation models for directly generating multi-frame multi-view image arrays. To alleviate potential conflicts between these two kinds of generation models, a variance-reducing sampling strategy is proposed. With the generated multi-frame multi-view image arrays, the proposed method introduces a dynamic and texture-decoupled
reconstruction strategy to reconstruct 4D videos. Experiments show that the proposed method can make use of these two kinds of generation models for 4D generation.

**Strengths:**

1. This proposed method takes advantage of existing multi-view 3D generation and video generation for 4D generation and does not require extra 4D training data.
2. Through the analysis, the score composition for denoising is reasonable to achieve the above goal.
3. Considering the potential conflicts between two kinds of generation models, the proposed method introduces variance-reducing sampling to reconcile the distinct modes at a higher noise level.

**Weaknesses:**

1. Since the proposed method leverages both multi-view 3D generation models and video generation models, maybe a reasonable baseline is to generate a video first via video generation models, then generate multi-view images based on the frames in this video using the multi-view 3D generation models. Finally, 4D reconstruction methods, such as [1], can be applied to generate 4D contents. It will be better to use this baseline to highlight the effectiveness of the proposed score composition.
[1] Li, Zhan, et al. "Spacetime gaussian feature splatting for real-time dynamic view synthesis." Proceedings of the IEEE/CVF Conference on Computer Vision and Pattern Recognition. 2024.
2. The experiments are weird. In Figure 4, the qualitative results for the stage-1 and stage-2 are almost the same. As mentioned in L524-525, the images generated in the first stage will show blurring details. Therefore, why are the results in two stage almost the same?
3. For the ablation studies, it is better to show quantitative results like Table 1 and Table 2.

**Questions:**

1. Can you try the baseline mentioned in W1 to further demonstrate the effectiveness of the proposed method?
2. Can the proposed method generalize to other multi-view 3D generation and video generation methods?
3. Can you show the time consumption for stage-1 and stage-2?
4. Is it possible for the proposed method to generate 4D contents with multiple objects?

---

> ### Author Response · Authors · 2024-11-22
> **Response to Reviewer AvRx  (Part 1)**
>
> We thank the reviewer for the detailed review as well as the suggestions for improvement. Our responses to the reviewer’s concerns are below:
>
> **Q1: Comparison with the baseline mentioned in W1.**
>
> Great suggestion. In ablation experiments of the main paper and supplementary video, we have compared with a similar baseline that only utilizes SV3D [2] for each frame.
> To ensure a fair evaluation of the improvements brought by score composition, we also incorporated variance-reducing sampling (VRS) into it.
> We further provide the quantitative comparison in the table below (also in Table 3 of the revised paper) with the suggested baseline (reconstructing images from only multi-view 3D generator with [5]):
>
> |  **Method**  | **LPIPS↓** | **CLIP(%)↑** | **FVD↓** |
> |  ----  | ----  |  ----  | ----  |
> | Baseline  | 14.86 | 89.61  | 1412.00 |
> | Ours  | 12.35 | 92.20  | 831.32 |
>
> The qualitative results from only the multi-view 3D generation model without VRS are shown **in the 00:31~00:35 of revised supplementary video**.
> It can be observed that the generated videos exhibit significant temporal inconsistencies due to the inherent stochasticity in the diffusion denoising process, making it challenging to be reconstructed into meaningful 4D assets, even with state-of-the-art 4D reconstruction methods [5] with superior fitting capability.
>
> **Q2: Similar results in two stages.**
>
> Thanks for the careful observation. Note that the second stage is a reconstruction stage, the almost identical results are partly ascribed to the Gaussian’s strong fitting ability for the training views. It also corroborates the high consistency of the images generated in the first stage. Besides, the multi-view references of examples in Figure 4 already have textureless back views (as shown **in Figure 8 of the revised appendix**), so the first-stage results usually do not exhibit significantly more blurry details in these views.
>
> **Q3: Quantitative results for ablation studies.**
>
> Thanks for the suggestion. We conduct quantitative ablations on the Consistent4D dataset, the results are presented in **Table 3 of revised appendix** and below:
>
> |  **VRS**  | **Score comp.**  | **Feat. blend**  | **LPIPS↓** | **CLIP(%)↑** | **FVD↓** |
> |  :----:  | :----:  |  :----:  | ----  | ----  | ----  |
> |  |  |  | 11.63 | 94.82  | 1248.49 |
> | √ |  |  | 12.79 | 91.85  | 1050.68 |
> | √ |  | √ | 12.65 | 92.18  | 898.66 |
> |  | √ |  | 11.86 | 94.30  | 1170.96 |
> | √ | √ |  | 12.35 | 92.20  | 831.32 |
>
> It aligns with our observations drawn from the qualitative ablations, where our result shines in video-level metric of FID due to its better temporal coherence.
>
> In the submitted version, we following common practice [3,4], only report qualitative ablations, as we believe they are sufficient to demonstrate the characteristics and differences from these baselines.
> More importantly, there is a lack of comprehensive quantitative metrics to evaluate the quality of generated assets in such tasks.
> Besides, we modified the evaluation of FVD in the revised paper by aligning the background color. While the comparison between previous metrics can also reflect the severity of temporal artifacts in the foreground object, the new results help to better align the scale with those reported in other 4D generation works.

---

> ### Author Response · Authors · 2024-11-22
> **Response to Reviewer AvRx (Part 2)**
>
> **Q4: Can the proposed method generalize to other multi-view 3D generation and video generation methods?**
>
> Yes, as long as they are diffusion-based generative models and share the same latent space.
> Our framework adopts an **architecture-agnostic** approach to combine heterogeneous scores from multi-view and video diffusion models.
>
> **Q5: Time consumption for stage-1 and stage-2.**
>
> We have already reported the total generation time in **Table 2** and indicated the time spent in the first stage at **L469** in the initial submission. Now we present them in the table below for more clear illustration.
>
> |    | Time |
> |  ----  | ----  |
> | Stage-1  | 8m31s |
> | Stage-2  | 3m33s |
> | Overall  | 12m4s |
>
> **Q6: Extending to multi objects.**
>
> Good question. The multi-view generator used in our framework is trained on object-centric data, limiting its generation to scenes containing a single object. Constrained by this, we limit the scope of our method to the generation of single dynamic objects, similar to prior 4D generation works [1].
> Usually, 4D results of multiple objects can be created by composing multiple dynamic objects generated independently.
> To better answer this question, we additionally provide an example of a composed scene with multiple dynamic objects **in the revised supplementary video (01:10~01:18)**, demonstrating scalability and generalization ability of our approach.
>
> > [1] Jiang, Yanqin, et al. Consistent4d: Consistent 360° dynamic object generation from monocular video. *ICLR*, 2024.
>
> > [2] Voleti, Vikram, et al. Sv3d: Novel multi-view synthesis and 3d generation from a single image using latent video diffusion. *ECCV*, 2024.
>
> > [3] Tang, Jiaxiang, et al. Dreamgaussian: Generative gaussian splatting for efficient 3d content creation. *ICLR*, 2024.
>
> > [4] Ren, Jiawei, et al. Dreamgaussian4d: Generative 4d gaussian splatting. *arXiv preprint*, 2023.
>
> > [5] Li, Zhan, et al. Spacetime gaussian feature splatting for real-time dynamic view synthesis. *CVPR*. 2024.

---

> > ### Comment · Reviewer_AvRx · 2024-11-26
> >
> > Thanks for your response. I still have some questions.
> >
> > As a 'training-free' or 'plug-in' module, I cannot see the detailed generalization examples of the proposed method to other multi-view 3D generation and video generation methods. This cannot support the claim in the paper.
> >
> > For Q3, the quantitative ablation for the decoupled reconstruction is not provided. I still think that limited qualitative results cannot fully support the effectiveness of different designs. Moreover, the table results in Q3 are different from those in Appendix, please correct them.
> >
> > For L484 and L505, the titles for two paragraphs are same, please correct them.

---

> > > ### Comment · Reviewer_UTNE · 2024-11-26
> > >
> > > "As a 'training-free' or 'plug-in' module, I cannot see the detailed generalization examples of the proposed method to other multi-view 3D generation and video generation methods. This cannot support the claim in the paper."
> > >
> > > I think it indeed needs a little more examples with various multi-view diffusers and VSDs in the revision. Since multi-view diffusers often shares a similar input and output patterns(from Zero123, SyncDreamer, Wonder3D to SV3D) and video diffusion models often shares a similar Latent_UNet(SVD, I2VGen-XL) or Latent_DiT(OpenSora-Plan, ViDU, Sora, CogVideo-X) structures, it is easy to extend Diffusion^2 to these methods. Despite the easiness, different multi-view diffusers may produce different views and with different modality(like normal map in wonder3d) and different VSD may have various condition mechanism, so demonstrating the generalization can also take massive efforts. So in my opinion, it is not very practical in the review stage.

---

> > > ### Comment · Reviewer_UTNE · 2024-11-26
> > >
> > > Besides, many follow-up works actually demonstrate the generalization like Vidu4D, SV4D, and Diffusion4D

---

> ### Author Response · Authors · 2024-11-25
> **Looking forward to your feedback**
>
> Dear Reviewer AvRx,
>
> We sincerely appreciate the reviewer's time for reviewing, and we really want to have a further discussion with the reviewer to see if our detailed explanations and additional results solves the concerns. We have addressed all the thoughtful questions raised by the reviewer *(eg, results, time consumption, extending to multi objects)* and we hope that our work’s contribution and impact are better highlighted with our responses. As the discussion phase is nearing its end, it would be great if the reviewer can kindly check our responses and provide feedback with further questions/concerns (if any). We would be more than happy to address them. Thank you!
>
> Best wishes,
>
> Authors

---

> ### Author Response · Authors · 2024-11-26
>
> Thanks to the reviewers for the time and effort dedicated to thoroughly reviewing our work.
>
> **As a 'training-free' or 'plug-in' module, I cannot see the detailed generalization examples of the proposed method to other multi-view 3D generation and video generation methods. This cannot support the claim in the paper.**
>
> Thanks for the comments. We first clarify the following facets:
>
> (i) For the 'training-free' property, our method indeed only utilizes the off-the-shelf multi-view image generative model and the video generative model without additional training.
>
> (ii) Although we did not claim the generalization ability implied by "plug-in" as our contribution in the paper for rigorousness, this property is evident in logic, as the proposed pipeline holds no bias on the model architecture. Therefore, we give an affirmative response to Q4.
>
> Considering the above points, we use SV3D [4] as the multi-view 3D generator given its superior performance than other MVDiffusions.
> It is sufficient to demonstrate the feasibility of our proposed approach.
> However, to solve the reviewer's concern, we additionally conducted experiments on an alternative multi-view generative model V3D [3] to empirically demonstrate our framework's applicability on other diffusion models.
> Due to the time constraint, we test this configuration on two cases.
> The results (`v3d.mp4`) are included in the supplementary material.
> Despite the slight Janus problem resulting from the limited capability of the adopted multi-view generative model, the generated 4D assets still exhibit overall spatial-temporal coherence, with significantly better consistency than that of the multi-view generative model-only baseline.
> This result further demonstrates the potential of our approach to scaling with various foundational diffusion models.
>
>
>
> **For Q3, the quantitative ablation for the decoupled reconstruction is not provided. I still think that limited qualitative results cannot fully support the effectiveness of different designs.**
>
> Actually, there is a lack of comprehensive quantitative metrics to evaluate the quality of generated assets in this task, so we following the widely recognized works [1,2] by providing the qualitative ablations in our initial submission, which can clearly illustrate our advantage against the baselines.
> In addition, we have also provided comprehensive quantitative ablations for the design of our first stage (which is the core part of the proposed framework) in our last response to Reviewer AvRx's Q3 (Quantitative results for ablation studies).
> Now we additionally included the ablations on the decoupled reconstruction in Table 3 of the revised appendix.
> From Table 3 (Appendix), we can observe that our full model (f) outperforms the baseline (g) without decoupled reconstruction.
>
> **Moreover, the table results in Q3 are different from those inthe  Appendix, please correct them.**
>
> Thanks for the eagle eyes! Actually, the results in Q3 and those in the Appendix (Table 3) are essentially the same except for the format of CLIP score. The results in the response of Q3 were reported as percentages omitting the "%", while the numbers in the CLIP score column of Table 3 (Appendix) are presented in the actual decimal form. For example, 94.82 in Q3 and 0.948 in Table 3. We have fix it in the response of Q3.
>
> **For L484 and L505, the titles for two paragraphs are same, please correct them.**
>
> Thanks for pointing out this typo, we have fix it.
>
> Hope our responses clarify the questions above. The reviewer’s constructive comments have been instrumental in guiding these refinements, and we are sincerely grateful for this guidance.
>
> > [1] Tang, Jiaxiang, et al. Dreamgaussian: Generative gaussian splatting for efficient 3d content creation. *ICLR*, 2024.
>
> > [2] Ren, Jiawei, et al. Dreamgaussian4d: Generative 4d gaussian splatting. *arXiv preprint*, 2023.
>
> > [3] Chen, Zilong, et al. V3d: Video diffusion models are effective 3d generators. *arXiv preprint*, 2024.
>
> > [4] Voleti, Vikram, et al. Sv3d: Novel multi-view synthesis and 3d generation from a single image using latent video diffusion. *ECCV*, 2024.

---

> ### Author Response · Authors · 2024-12-01
>
> Dear Reviewer AvRx,
>
> We sincerely thank the reviewer for the valuable time on reviewing and discussion. As the discussion phase is nearing its end, we wondered if the reviewer might still have any concerns that we could address. After the thorough discussion with the reviewer, we believe our response on *more experiments on different multi-view 3D generator, time consumption, extending to multi objects, and typos* have addressed all thoughtful questions raised by the reviewer, and hope these results helps the final recommendation. We genuinely appreciate the reviewer’s valuable suggestion for improving our work!
>
> Best wishes,
>
> Authors

---

> > ### Comment · Reviewer_AvRx · 2024-12-02
> >
> > Thank you very much for your detailed response, my concerns have been addressed. I think that this work provides an effective and resource-friendly alternative for 4D generation, I will raise my score! In addition, may I ask how many cases are used in the ablation study?

---

> > > ### Author Response · Authors · 2024-12-02
> > >
> > > We would like to express our sincere gratitude to Reviewer AvRx for the open-mind and great contributions for improving this work! In the quantitative ablation study, each setting is evaluated across all seven samples in the commonly used Consistent4D test dataset, which are synthesized from animated objects.

---

> > > > ### Comment · Reviewer_AvRx · 2024-12-02
> > > >
> > > > Thanks for your prompt reply. It is better to show these ablation details in the revised version.

---

> > > > > ### Author Response · Authors · 2024-12-02
> > > > >
> > > > > Certainly, we will include them in the revised version! Thanks once again for the detailed review and kind reminder.

---

### Official Review · Reviewer_s4pt · 2024-11-03

**Soundness:** 2
**Presentation:** 2
**Contribution:** 2
**Rating:** 5
**Confidence:** 3

**Summary:**

This paper presents Diffusion^2, an interesting framework that aims to combine video and multi-view diffusion models through score composition for 4D content generation. The method proposes parallel generation of multi-view videos without requiring synchronized 4D training data, offering a novel theoretical perspective on combining different types of diffusion priors.

**Strengths:**

- Innovative conceptual approach to leveraging video and multi-view diffusion priors
- Well-formulated theoretical framework for score composition
- Thoughtful parallel generation architecture design
- Clear and thorough ablation studies

**Weaknesses:**

The method has some practicality issues:
- Requires 8-16 A6000 GPUs for basic operation
- Still takes 20 minutes per generation even with massive compute
- Memory-speed tradeoff makes it impractical for most real applications
It would be good to hear the authors discussing about the path to reduce these extreme hardware requirements.


The experimental validation could be more comprehensive:
- Most demonstrations focus on relatively simple motions
- Additional results on more complex scenarios (e.g., rotational movements, articulated motions) would help better understand the method's capabilities
- Further exploration of challenging cases would strengthen the evaluation.

**Questions:**

- Have the authors considered strategies to reduce the computational requirements?
- Could the authors provide insights on how the method might handle more complex motion patterns?
- What are the main challenges in extending this approach to more diverse scenarios?

---

> ### Author Response · Authors · 2024-11-22
> **Response to Reviewer s4pt**
>
> We thank the reviewer for the acknowledgment of the innovation of our work as well as the valuable suggestions for improvement. Our responses to the reviewer’s comments are below:
>
> **Q1: Requires 8-16 A6000 GPUs for basic operation？**
>
> This is a constructive question regarding the application of our Diffusion$^2$. We first clarify the following facets:
>
> 1. **The use of 8~16 A6000 GPUs for inference is not mandatory.** We adopt it as a setting to demonstrate the advantage on maximum inference efficiency benefited from high parallelism, which is more important for the online tasks.
> 2. As mentioned in L469, multi-GPU is only used in the first stage, which takes less than 10 minutes. In addition, if we sacrifice parallelism and only use a **single GPU**, our framework is still **significantly faster than representative SDS-based methods** (e.g. 2x than Consistent4D [1]). We depict this trade-off **in Figure 7 of the revised appendix.**
>
> Actually, even without considering parallelism, our approach almost constitutes a lower bound on the time required for sampling $V(=21) \times F(=25)$ images with diffusion models to reconstruct 4D asset: we only perform the denoising process twice for each image without iterative refinement. However, sampling such a dense set of multi-view, multi-frame images may be unnecessary for reconstructing 4D representations. So the further reduction in the computational requirements lies in the redundancy of such a dense image array. Now we sampled 21 views for each frame to leverage an existing multi-view generator [2]. In practical applications, fine-tuning this generator to support a smaller $V=8$ (as in concurrent work [4]) may not substantially impact performance but can immediately reduce computational requirements by 2/3.
>
> Besides, in real applications, some efforts on engineering optimizations can also be made to mitigate the efficiency bottleneck introduced by multi-GPU synchronization and communication revealed in Figure 7 of the revised appendix.
>
> **Q2: Further exploration of challenging cases.**
>
> Great suggestion. The examples we currently use are primarily sourced from commonly used public 4D generation datasets [1,3]. Among them, the example *sighing frog* (shown in Figure 4 and supplementary video) retains the largest motion within the Consistent4D dataset, thus posing significant challenges for many prior methods. Both SDS-based and feature blending-based approaches struggle to establish the association between frames with substantial differences. In contrast, our proposed method demonstrates the potential to handle complex motions with global receptive fields of two score estimators.
>
> For 4D objects with rotational movements or articulated motions which pose distinct challenges, We show additional examples **in the revised supplementary video (01:04~01:08)** to provide more insights and demonstrate the effectiveness of our approach.
>
> **Q3: Extending to more diverse scenarios.**
>
> The capability of the proposed framework is mainly constrained by the foundational diffusion model. Since the multi-view generator [2] used in this work focuses on object-centric data, we limit our exploration to the generation of single dynamic objects, following existing works in this field [1,3,4].
>
> For more diverse 4D scenarios, as long as the two diffusion models share a same latent space to model their marginal distributions, a similar sampling approach can be applied.
>
> > [1] Jiang, Yanqin, et al. Consistent4d: Consistent 360° dynamic object generation from monocular video. *ICLR*, 2024.
>
> > [2] Voleti, Vikram, et al. Sv3d: Novel multi-view synthesis and 3d generation from a single image using latent video diffusion. *ECCV*, 2024.
>
> > [3] Zhao, Yuyang, et al. Animate124: Animating one image to 4d dynamic scene. *arXiv preprint*, 2023.
>
> > [4] Xie, Yiming, et al. Sv4d: Dynamic 3d content generation with multi-frame and multi-view consistency. *arXiv preprint*, 2024.

---

> > ### Author Response · Authors · 2024-12-01
> >
> > Dear Reviewer s4pt
> >
> > We sincerely thank the reviewer for the valuable comments and suggestions. As the discussion phase is nearing its end, we wondered if the reviewer might still have any concerns that we could address. We believe our responses on *computational requirements, extending to more challenging cases* addressed all the questions/concerns, and hope our response helps the final recommendation. Thank you!
> >
> > Best wishes,
> >
> > Authors

---

> ### Author Response · Authors · 2024-11-25
> **Looking forward to your feedback**
>
> Dear Reviewer s4pt,
>
> We sincerely appreciate the reviewer's time for reviewing, and we really want to have a further discussion with the reviewer to see if our detailed explanations and additional results solve the concerns. We have addressed all the thoughtful questions raised by the reviewer *(eg, computational requirements, extending to more challenging cases)* and we hope that our work’s contribution and impact are better highlighted with our responses. As the discussion phase is nearing its end, it would be great if the reviewer can kindly check our responses and provide feedback with further questions/concerns (if any). We would be more than happy to address them. Thank you!
>
> Best wishes,
>
> Authors

---

### Comment · Area_Chair_ztSV · 2024-11-25
**Please read rebuttal and reply if you have further questions**

Dear Reviewers,

Thanks again for serving for ICLR, the discussion period between authors and reviewers is approaching (November 27 at 11:59pm AoE), please read the rebuttal and ask questions if you have any. Your timely response is important and highly appreciated.

Thanks,
AC

---

### Meta-Review · Area_Chair_ztSV · 2024-12-16

**Metareview:**

This paper proposes a method to align the latent space of MV diffusion and video diffusion models for 4D video generation. The main idea is based on a conditional independent assumption and designed Score Distillation process. The developed algorithm can generate 4D video in a relatively efficient manner and present reasonable results. During rebuttal, reviewers raised questions about efficiency, missing baselines, ablation studies, most of them are addressed during the rebuttal. Given that the method can be a new milestone for efficient 4D generation and contribute to future related researches, the paper is recommended for acceptance.

**Additional Comments On Reviewer Discussion:**

The main points raised by reviewers include
- Efficiency (Reviewer s4pt)
- Performance (Reviewer rKU7)
- Missing baselines (Reviewer AvRx)
- Missing ablations (Reviewer AvRx)
- Other questions about the assumption and technical details (Reviewer UTNE)

The authors have responded to all of these concerns with sufficient supporting materials. All reviewers (except Reviewer s4pt) claimed that most of their concerns are addressed.

---

### Decision · Program_Chairs · 2025-01-22

Accept (Poster)